# MADG: Margin-based Adversarial Learning for Domain Generalization

**Aveen Dayal**
Indian Institute of Technology Hyderabad
ai21resch11003@iith.ac.in

**Vimal K B**
Indian Institute of Technology Hyderabad
vimalkb96@gmail.com

**Linga Reddy Cenkeramaddi**
University of Agder
linga.cenkeramaddi@uia.no

**C Krishna Mohan**
Indian Institute of Technology Hyderabad
ckm@cse.iith.ac.in

**Abhinav Kumar**
Indian Institute of Technology Hyderabad
abhinavkumar@ee.iith.ac.in

**Vineeth N Balasubramanian**
Indian Institute of Technology Hyderabad
vineethnb@cse.iith.ac.in

## Abstract

Domain Generalization (DG) techniques have emerged as a popular approach to address the challenges of domain shift in Deep Learning (DL), with the goal of generalizing well to the target domain unseen during the training. In recent years, numerous methods have been proposed to address the DG setting, among which one popular approach is the adversarial learning-based methodology. The main idea behind adversarial DG methods is to learn domain-invariant features by minimizing a discrepancy metric. However, most adversarial DG methods use 0-1 loss based $\mathcal{H}\Delta\mathcal{H}$ divergence metric. In contrast, the margin loss-based discrepancy metric has the following advantages: more informative, tighter, practical, and efficiently optimizable. To mitigate this gap, this work proposes a novel adversarial learning DG algorithm, **MADG**, motivated by a margin loss-based discrepancy metric. The proposed **MADG** model learns domain-invariant features across all source domains and uses adversarial training to generalize well to the unseen target domain. We also provide a theoretical analysis of the proposed **MADG** model based on the unseen target error bound. Specifically, we construct the link between the source and unseen domains in the real-valued hypothesis space and derive the generalization bound using margin loss and Rademacher complexity. We extensively experiment with the **MADG** model on popular real-world DG datasets, VLCS, PACS, OfficeHome, DomainNet, and TerraIncognita. We evaluate the proposed algorithm on DomainBed's benchmark and observe consistent performance across all the datasets.

## 1  Introduction

Over the past decade, Deep Neural Networks (DNNs) have demonstrated exceptional performance across various fields, including robotics [1], medical imaging [2], agriculture [3], and more. However, the effectiveness of DNNs in supervised learning environments relies heavily on the independently and identically distributed ($i.i.d.$) assumption of the training and test (target) data. Unfortunately, in reality, this assumption can be compromised due to domain shifts in target data [4]. For example, a model trained on different domains of image data may perform poorly when presented with an image of a known label featuring an unseen background or viewpoint, as explained in [5]. To address this issue, researchers have developed techniques under the framework of domain adaptation (DA) [6]. The key idea behind DA is to adapt a model trained on a source dataset to minimize the generalization

37th Conference on Neural Information Processing Systems (NeurIPS 2023).

error on the target dataset [7]. However, the major limitation of DA is that the target data, whether labeled or unlabeled, must be available during training. In contrast, the Domain Generalization (DG) setting aims to leverage knowledge from similar domains to classify previously unseen domains [8].

Recent years have seen the development of various algorithms addressing the DG setting, among which one popular approach is the adversarial learning-based methodology [9–14]. Other approaches can be broadly classified into meta-learning techniques [15–17], data augmentation methods [18–20], self-supervised learning methods [21–23], regularization-based methods [24–29], and so on. Additionally, there have been concerted efforts towards the development of benchmark datasets [30–35] for the DG setting to study methods. The assumption of distribution shift, i.e., smooth variation between the conditional distribution $\mathbb{P}(Y|X)$ and the marginal distribution $\mathbb{P}(X)$ is also common in DG literature [36; 37].

The main objective of DG methods is to model the functional relationship between the input space $\mathcal{X}$ and the label space $\mathcal{Y}$ for all domains. To achieve this, adversarial DG methods learn a domain-invariant representation by minimizing a discrepancy metric among the source domains. However, despite the myriad efforts, most adversarial learning methods use the 0-1 loss based $\mathcal{H}\Delta\mathcal{H}$ divergence metric [38] for domain alignment. In contrast, divergence metrics based on margin loss are more informative [39], practical, and efficiently optimizable [40]. This work addresses this gap by proposing a novel adversarial DG algorithm, **MADG**, motivated by a margin-based divergence metric. The proposed **MADG** leverages margin-based disparity discrepancy [40] to estimate source domain discrepancies in the DG setting and uses adversarial training to ensure that **MADG** generalizes well to unseen target domains. We also theoretically analyze the proposed **MADG** algorithm based on bounds for the unseen target error. The proposed generalization bound uses the Rademacher complexity framework [41], which provides data-dependent estimates of functional class complexities. The effectiveness of the proposed algorithm is demonstrated through extensive experiments on multiple benchmark DG datasets.

The key contributions of this work can be summarized as follows: (i) We introduce the use of margin loss and a corresponding scoring function to formulate the relationship between domains and develop upper bounds for the unseen target error (the first such margin-based effort in the DG setting, to the best of our knowledge); (ii) We subsequently analyze the generalization bound in terms of functional class complexity using the Rademacher complexity framework; (iii) We propose a novel margin-based adversarial DG training algorithm, **MADG**, motivated by our theoretical results; and (iv) We study the proposed method on five well-known benchmark datasets in the DomainBed setup, providing a higher average accuracy and consistency in model performance across these datasets.

## 2    Related Work

In this section, we discuss earlier literature proposed specifically for adversarial DG, as well as theoretical analysis for the DG problem in general. We discuss other DG literature across a broader set of categories in the Appendix. The main idea behind existing adversarial DG methods is to minimize the $\mathcal{H}\Delta\mathcal{H}$ divergence by employing a minimax optimization between a domain discriminator and a classifier to learn domain-invariant features. One of the early works [9] proposed a method that iteratively divided samples into latent domains via clustering and trained the domain-invariant feature extractor via adversarial learning. Other efforts extended such an approach of adversarial learning with different divergence metrics and regularization techniques. Lin et al. [10] proposed a multi-dataset feature generalization network (MMFA-AAE) based on an adversarial auto-encoder to learn a generalized domain-invariant latent feature representation with the Maximum Mean Discrepancy (MMD) measure to align distributions across multiple domains. Deng et al. [11] examined adversarial censoring techniques to learn invariant representations from multiple domains. Zhao et al. [12] proposed an entropy regularization term along with adversarial loss to ensure conditional invariance of learned features. Akuzawa et al. [13] proposed the notion of accuracy-constrained domain invariance, and then developed an adversarial feature learning method with accuracy constraint (AFLAC), which explicitly provided invariance on adversarial training. Rahman et al. [42] proposed a correlation-aware adversarial DG framework where the features of the source and target data are minimized using correlation alignment along with adversarial learning. All these prior works on adversarial DG methods use a 0-1 loss based $\mathcal{H}\Delta\mathcal{H}$ discrepancy metric to align source domains. In this work, we instead leverage a margin-based disparity discrepancy and propose a new adversarial learning strategy founded on our theoretical analysis. The margin loss is advantageous compared to the 0-1 loss as it provides informative generalization bounds, tightness, classifier-aware alignment, and efficient optimization. We discuss each of these advantages in detail in Section 4.

Early efforts for theoretical analysis for the DG problem [43; 44] used kernel-based approaches to the problem setting with corresponding generalization error analysis, and showed empirical results using traditional machine learning methods. A few other efforts [45–47] also followed a similar path, and focused on kernel-based approaches in traditional methods. From a deep learning perspective, an early attempt at such a theoretically motivated method was proposed in [48] using a convex hull formulation and distribution matching to obtain generalization bounds, which also is an inspiration for parts of our work. In [49], the authors study theoretical bounds in the DG setting using three new concepts: feature invariance across domains, feature discriminability for the classification task, and a feature expansion function to generalize the first two concepts from the source to the target domain. [50] identified measures relating to the Fisher information, predictive entropy, and maximum mean discrepancy to be good predictors of out-of-distribution generalization of empirical risk minimization (ERM) models in the DG setting. [24] proposed generalization bounds in terms of the model's Rademacher complexity and suggested using regularized ERM models for the DG problem. Finally, [14] introduced an online game in which one model (player) reduces the error on test data provided by an adversarial player, but do not provide any empirical studies. In this work, we draw inspiration from [48] by considering a convex combination of source domains and leverage this to propose a new margin-based approach to the theoretical analysis of DG, which helps develop an informative generalization bound. Motivated by this analysis, we propose the **MADG** algorithm, a novel margin-based adversarial learning approach for DG, which shows consistent performance over other state-of-the-art methods across benchmarks.

## 3    Preliminaries

We consider a set of source domains $\mathcal{D}_{\mathcal{S}_i}$ with $i \in \{1, 2, \ldots, N_s\}$ and a set of unseen domains $\mathcal{D}_{U_m}$ with $m \in \{1, 2, \ldots, N_u\}$. We use $\mathcal{D}$ to refer to any of these domains when the index is not relevant. We use the terms 'domain' and 'distribution' interchangeably in this work. Each such domain $\mathcal{D} \subseteq \mathcal{X} \times \mathcal{Y}$, where $\mathcal{X}$ is an input space and $\mathcal{Y}$ is an output space, which is $\{0, 1\}$ in binary classification and $\{1, \cdots, k\}$ in multi-class classification. We use $\hat{D}$ to denote a set of samples drawn independently from $\mathcal{D}$, i.e. $\hat{D} = \{(x_i, y_i)\}_{i=1}^n$ where $x_i \in \mathcal{X}$ and $y_i \in \mathcal{Y}, \forall i \in \{1, 2, \ldots, n\}$. We use $(x, y)$ to refer to a labeled sample $(x_i, y_i)$, when the index is not relevant.

As in [51], we consider the multi-class setting with a hypothesis space $\mathcal{F}$ of scoring functions $f : \mathcal{X} \to \mathbb{R}^{|\mathcal{Y}|} = \mathbb{R}^k$, where the outputs report the confidence of the prediction on each dimension. Similar to [40], we use $f(x, y)$ to denote the component of $f(x)$ that corresponds to the label $y$. In order to obtain the final predicted label, we also consider a labeling function space $\mathcal{H}$ containing $h_f : \mathcal{X} \to \mathcal{Y}$ such that $h_f(x) = \mathrm{argmax}_{y \in \mathcal{Y}} f(x, y)$, i.e. the predicted label assigned to data sample $x$ is the one resulting in the largest confidence score.

The expected error rate of a classifier $h \in \mathcal{H}$ with respect to a distribution $\mathcal{D}$ is defined as $err_{\mathcal{D}}(h) \triangleq \mathbb{E}_{(x,y) \sim \mathcal{D}} \mathbb{1}[h(x) \neq y]$ where $\mathbb{1}$ is the indicator function. We also define the margin $\rho_f(\cdot)$ and the corresponding margin loss of hypothesis $f$ for a labeled example $(x, y)$ as follows:

$$\rho_f(x, y) \triangleq \frac{1}{2}\Big(f(x, y) - \max_{y' \neq y} f(x, y')\Big); \qquad err_{\mathcal{D}}^{(\rho)}(f) \triangleq \mathbb{E}_{x \sim \mathcal{D}}\big[\Phi_\rho \circ \rho_f(x, y)\big] \qquad (1)$$

where $\circ$ denotes the composition function and $\Phi_\rho$ is:

$$\Phi_\rho(t) = \begin{cases} 0 & \text{if } \rho \leq t \\ 1 - \frac{t}{\rho} & \text{if } 0 \leq t \leq \rho \\ 1 & \text{if } t \leq 0 \end{cases} \qquad (2)$$

The margin disparity and its empirical version are then defined as:

$$\mathrm{disp}_{\mathcal{D}}^{(\rho)}\big(f', f\big) \triangleq \mathbb{E}_{\mathcal{D}}\big[\Phi_\rho \circ \rho_{f'}(., h_f)\big] \qquad (3)$$

$$\mathrm{disp}_{\hat{D}}^{(\rho)}\big(f', f\big) \triangleq \mathbb{E}_{\hat{D}}\big[\Phi_\rho \circ \rho_{f'}(., h_f)\big] = \frac{1}{n}\sum_{i=1}^n \Phi_\rho \circ \rho_{f'}\big(x_i, h_f(x_i)\big) \qquad (4)$$

where $f$ and $f'$ are different scoring functions. Margin disparity discrepancy (MDD) is used to quantify the degree of discrepancy/disagreement between decision boundaries of classifiers (using their margins) trained on different domains in our context. Such a measure can be used to evaluate the generalization performance of a model across multiple domains. The MDD and its empirical version are defined in Eqn. (5) and Eqn. (6) below, respectively.

$$d_{f,\mathcal{F}}^{(\rho)}\big(\mathcal{D}_{S_i}, \mathcal{D}_{S_k}\big) \triangleq \sup_{f' \in \mathcal{F}} \Big(\mathrm{disp}_{\mathcal{D}_{S_k}}^{(\rho)}\big(f', f\big) - \mathrm{disp}_{\mathcal{D}_{S_i}}^{(\rho)}\big(f', f\big)\Big) \tag{5}$$

$$d_{f,\mathcal{F}}^{(\rho)}\big(\hat{D}_{S_i}, \hat{D}_{S_k}\big) \triangleq \sup_{f' \in \mathcal{F}} \Big(\mathrm{disp}_{\hat{D}_{S_k}}^{(\rho)}\big(f', f\big) - \mathrm{disp}_{\hat{D}_{S_i}}^{(\rho)}\big(f', f\big)\Big) \tag{6}$$

We use the above terminologies from [40], which however focused on domain adaptation (with one source domain and one seen target domain). Handling multiple seen domains and an unseen target domain in the DG setting is non-trivial, which we focus on in this work.

## 4 Margin-based Approach to Domain Generalization: Theory

This section presents the theoretical motivation for our margin-based approach to the Domain Generalization (DG) problem. Our approach is based on considering the margin disparity discrepancy (MDD), defined above, between the source domains, thereby obtaining a generalization bound. We believe such a margin-based approach has a few advantages: (i) *Informative:* Generalization bounds based on margin loss for classification provide more information (than 0-1 loss-based bounds) by establishing a dependency between any margin function satisfying the Lipschitz condition and the upper bounds, as demonstrated in well-known earlier work [39]. (ii) *Tightness:* In contrast to using a 0-1 loss to identify samples causing disagreement between classifiers, margin loss computes disagreement between scoring functions using a smooth function parameterized by the threshold ($\rho$). Consequently, the number of samples causing agreement between classifiers is less, resulting in such an MDD-based bound being tighter, as shown in Fig. 1. (iii) *Classifier-aware Discrepancy:* MDD considers the classifier function while measuring discrepancy; as shown in Eqns 5 and 6, the supremum is computed over $f'$ while holding $f$ constant ($f$ learns the posterior distribution $\mathbb{P}(y|x)$ discriminatively for the classification task). This provides a classifier-aware approach to computing discrepancy.

(iv) *Efficient Optimization and Practicality:* The definition of MDD, as in Eqn. (5), only requires taking the supremum over one hypothesis. Therefore, compared to other divergence measures such as $\mathcal{H}\Delta\mathcal{H}$ [38], MDD can be optimized with ease and is practically useful, as also stated in [40].

In this section, we derive a generalization bound for an unseen domain based on the margin-based MDD loss in the DG setting. To this end, we first show an upper bound on the unlabeled source domain error given other labeled source domains (Lemma 1 and Theorem 1). We then leverage this to develop the upper bound for the error on an unseen domain that is not necessarily a source domain (Lemma 2, Theorem 2, and Corollary 1). We subsequently analyze the upper bound

Figure 1: Space of intersection (agreement) in MDD (yellow) is reduced as compared to 0-1 loss (blue + yellow) between $f$ and $f'$ for labels $\{0, 1\}$.

from Corollary 1 using the Rademacher complexity framework and develop our final generalization bound for the unseen target domain in the DG setting (Lemma 3 and Theorem 3) using our margin-based loss. In Sec 5, we show the formulation of the proposed adversarial learning algorithm, MADG, motivated by the generalization bound in Sec 4, that employs MDD to address the DG problem.

We begin by considering a setting where training data consists of $N_s - 1$ labeled source domains $\mathcal{D}_{S_i}, i = \{1, \ldots, N_s - 1\}$ and a single unlabeled source domain $\mathcal{D}_{S_T}$, and show an upper bound on error in this setting. We later use this to show an upper bound for the DG setting. We first establish an upper bound on the MDD between a weighted sum of the labeled source domains and the unlabeled one in Lemma 1 below.

**Lemma 1** *Consider a weighted sum of $N_s - 1$ labeled source distributions defined as $\mathcal{D}_{\bar{S}} := \sum_{i=1}^{N_s-1} \alpha_i \mathcal{D}_{S_i}$, where $\alpha_i$s are mixture co-efficients s.t. $\sum_{i=1}^{N_s-1} \alpha_i = 1$, and $f$ is a scoring function. Then*

$$d_{f,\mathcal{F}}^{(\rho)}\big(\mathcal{D}_{\bar{S}}, \mathcal{D}_{S_T}\big) \leq \sum_{i=1}^{N_s-1} \alpha_i d_{f,\mathcal{F}}^{(\rho)}\big(\mathcal{D}_{S_i}, \mathcal{D}_{S_T}\big) \tag{7}$$

Detailed proofs for all our theoretical results are provided in the Appendix. It follows from Lemma 1 that an effective way to minimize the discrepancy between the unlabeled source domain and the mixture of labeled source domains in the hypothesis space is by minimizing the convex sum of the pairwise MDD between each labeled and unlabeled source domain. Building upon this insight, we provide bounds on the unlabeled source error below in Theorem 1.

**Theorem 1** *Consider a scoring function $f$, unlabeled source domain $\mathcal{D}_{\mathcal{S}_T}$ and a mixture of $N_s -$ 1 source distributions denoted as $\mathcal{D}_{\bar{\mathcal{S}}} := \sum_{i=1}^{N_s-1} \alpha_i \mathcal{D}_{\mathcal{S}_i}$, where $\alpha_i s$ is mixture co-efficients s.t. $\sum_{i=1}^{N_s-1} \alpha_i = 1$. Then the error on the unlabeled source $\mathcal{D}_{\mathcal{S}_T}$ is bounded as:*

$$err_{\mathcal{D}_{\mathcal{S}_T}}(h_f) \leq \sum_{i=1}^{N_s-1} \alpha_i \left( err_{\mathcal{D}_{\mathcal{S}_i}}^{(\rho)}(f) + d_{f,\mathcal{F}}^{(\rho)}(\mathcal{D}_{\mathcal{S}_i}, \mathcal{D}_{\mathcal{S}_T}) \right) + \hat{\lambda} \tag{8}$$

**Remark 1:** From Theorem 1, we observe that the unlabeled source error is upper bounded by the labeled source errors, the pairwise discrepancy between each labeled and unlabeled source domain, and the ideal margin loss described below in Remark 2. We can also interpret Theorem 1 as an upper bound for the multi-source domain adaptation (DA) setting. While this is not our primary focus in this work, we report preliminary empirical results for a multi-source DA algorithm that reduces the first two terms in Theorem 1 in the Appendix.

**Remark 2:** The ideal loss $\hat{\lambda}$ is defined as $\hat{\lambda} = \min_{f^* \in \mathcal{F}} \left( \sum_{i=1}^{N_s-1} \alpha_i \, err_{\mathcal{D}_{\mathcal{S}_i}}^{(\rho)}(f^*) + err_{\mathcal{D}_{\mathcal{S}_T}}^{(\rho)}(f^*) \right)$ and is a constant that is independent of function $f$.

Building on Theorem 1, we now develop bounds for the DG setting, where all source domains ($N_s$) are labeled, and the target domain is unseen during training. To this end, we consider the relationship between the unseen target and multiple source domains. In particular, we derive our error bound on the unseen target domain using the convex hull of the labeled source domains and MDD. We also provide a DG algorithm motivated by our theoretical analysis, which we describe later in Sec 5.

Before stating our main theorem, we present Lemma 2, which states that if the maximum MDD between any two sources is bounded above by $\epsilon$, then the MDD between any two domains that belong to the convex hull of all the sources will also be bounded by $\epsilon$. Let $\Lambda_s$ be the convex hull of source domains defined as $\Lambda_s = \left\{ \bar{\mathcal{D}} : \bar{\mathcal{D}} = \sum_{i=1}^{N_s} \pi_i \mathcal{D}_{\mathcal{S}_i} \text{ where } \sum_{i=1}^{N_s} \pi_i = 1 \right\}$.

**Lemma 2** *Let $d_{f,\mathcal{F}}^{(\rho)}(\mathcal{D}_{\mathcal{S}_i}, \mathcal{D}_{\mathcal{S}_k}) \leq \epsilon \; \forall i, k \in \{1, 2, \dots, N_s\}$ and $f$ be a scoring function. Then the following inequality holds for the MDD between any pair of domains $\mathcal{D}', \mathcal{D}'' \in \Lambda_s$:*

$$d_{f,\mathcal{F}}^{(\rho)}(\mathcal{D}', \mathcal{D}'') \leq \epsilon \tag{9}$$

We observe from Lemma 2 that in the hypothesis space, the discrepancy among all the domains (seen or unseen) that belong to the convex hull $\Lambda_s$ can be reduced by minimizing the maximum MDD between two source domains. With the necessary tools at hand, we state the key theorem, i.e., the unseen target error bound for the DG problem below in Theorem 2.

**Theorem 2** *Consider a mixture of $N_s$ source distributions, scoring function $f$, unseen domain $\mathcal{D}_{U_m}$, and $\gamma = d_{f,\mathcal{F}}^{(\rho)}(\mathcal{D}_{U_m}, \bar{\mathcal{D}}_U)$ where $\bar{\mathcal{D}}_U$ is the projection of $\mathcal{D}_{U_m}$ onto the convex hull of the sources i.e. $\bar{\mathcal{D}}_U = argmin_{\pi_1, \pi_2, \dots, \pi_{N_s}} d_{f,\mathcal{F}}^{(\rho)} \left( \mathcal{D}_{U_m}, \sum_{i=1}^{N_s} \pi_i \mathcal{D}_{\mathcal{S}_i} \right), \sum_{i=1}^{N_s} \pi_i = 1$. Then, the unseen target error is bounded as follows:*

$$err_{\mathcal{D}_{U_m}}(h_f) \leq \sum_{i=1}^{N_s} \pi_i \left( err_{\mathcal{D}_{\mathcal{S}_i}}^{(\rho)}(f) \right) + \epsilon + \gamma + \bar{\lambda} \tag{10}$$

**Remark 3:** As defined in Theorem 2, $\gamma = d_{f,\mathcal{F}}^{(\rho)}(\mathcal{D}_{U_m}, \bar{\mathcal{D}}_U)$. Two scenarios therefore arise: $\gamma = 0$ when the unseen domain falls in the convex hull, i.e., $\mathcal{D}_{U_m} = \sum_{i=1}^{N_s} \pi_i \mathcal{D}_{\mathcal{S}_i}$ or $\gamma > 0$ when the unseen domain cannot be represented by available domains alone. Thus this parameter can be interpreted as the need for diverse source domains. The more diverse the source domains are, the smaller the value of $\gamma$.

**Remark 4:** As seen from Lemma 2, $\epsilon$ is defined as the upper bound for the MDD between any two domains that belong to the convex hull $\Lambda_s$ formed by the source domains. Thus, we can also interpret $\epsilon$ as the highest MDD value among the source domains as shown below in Eqn 11.

$$\epsilon = d_{f,\mathcal{F}}^{(\rho)}(\mathcal{D}_{\mathcal{S}_{i'}}, \mathcal{D}_{\mathcal{S}_{k'}}), \text{ s.t. } d_{f,\mathcal{F}}^{(\rho)}(\mathcal{D}_{\mathcal{S}_{i'}}, \mathcal{D}_{\mathcal{S}_{k'}}) \geq d_{f,\mathcal{F}}^{(\rho)}(\mathcal{D}_{\mathcal{S}_i}, \mathcal{D}_{\mathcal{S}_k}) \; \forall i, k, i', k' \in \{1, \dots, N_s\} \tag{11}$$

Equipped with this definition for $\epsilon$, we can re-state Theorem 2 as Corollary 1 below.

**Corollary 1** *Consider a mixture of $N_s$ source distributions, scoring function $f$, unseen domain $\mathcal{D}_{U_m}$, $\gamma = d_{f,\mathcal{F}}^{(\rho)}(\mathcal{D}_{U_m}, \bar{\mathcal{D}}_U)$, and $d_{f,\mathcal{F}}^{(\rho)}(\mathcal{D}_{\mathcal{S}_{i'}}, \mathcal{D}_{\mathcal{S}_{k'}})$ where $d_{f,\mathcal{F}}^{(\rho)}(\mathcal{D}_{\mathcal{S}_{i'}}, \mathcal{D}_{\mathcal{S}_{k'}}) \geq d_{f,\mathcal{F}}^{(\rho)}(\mathcal{D}_{\mathcal{S}_i}, \mathcal{D}_{\mathcal{S}_k}) \; \forall i, k, i', k' \in \{1, 2, \dots, N_s\}$. Then the unseen target error is bounded as follows:*

$$err_{\mathcal{D}_{U_m}}(h_f) \leq \sum_{i=1}^{N_s} \pi_i \Big( err_{\mathcal{D}_{\mathcal{S}_i}}^{(\rho)}(f) \Big) + d_{f,\mathcal{F}}^{(\rho)}\big(\mathcal{D}_{\mathcal{S}_{i'}}, \mathcal{D}_{\mathcal{S}_{k'}}\big) + \gamma + \bar{\lambda} \tag{12}$$

As seen from Corollary 1, the unseen target error is bounded above by the source errors, maximum MDD between two source domains, $\gamma$, and the ideal loss given by $\bar{\lambda} = \min_{f^* \in \mathcal{F}} \left( \sum_{i=1}^{N_s} \pi_i err_{\mathcal{D}_{\mathcal{S}_i}}^{(\rho)}(f^*) + err_{\mathcal{D}_{U_m}}^{(\rho)}(f^*) \right)$.

Before we derive the generalization bound with the Rademacher complexity framework, we define $\Pi_{\mathcal{H}}\mathcal{F} = \left\{ x \mapsto f\big(x, h(x)\big) \Big| h \in \mathcal{H}, f \in \mathcal{F} \right\}$ [40] and Rademacher complexity $\mathfrak{R}_{n,\mathcal{D}}$ in Definition 1.

**Definition 1** *Let $\mathcal{F}$ be a class of functions such that $f \in \mathcal{F} : \mathcal{X} \times \mathcal{Y} \to [a, b]$ and $\hat{D} = \big\{ (x_1, y_1) \ldots, (x_n, y_n) \big\}$ be a fixed sample of size $n$ drawn from $\mathcal{D}$ over $\mathcal{X} \times \mathcal{Y}$. Then the Rademacher complexity of $\mathcal{F}$ is defined as:*

$$\mathfrak{R}_{n,\mathcal{D}}(\mathcal{F}) \triangleq \mathbb{E}_{\hat{D} \sim \mathcal{D}^n} \left[ \mathbb{E}_\sigma \left[ \sup_{f \in \mathcal{F}} \frac{1}{n} \sum_{i=1}^{n} \sigma_i f(x_i, y_i) \right] \right] \tag{13}$$

where $\sigma_i$s are independent Rademacher variables that assume values in $\{-1, +1\}$. We now leverage Lemma 3.6 in [40] to derive our Lemma 3.

**Lemma 3** *For any $\delta > 0$, with probability $1 - 2\delta$, the following holds simultaneously for any scoring function $f$:*

$$\left| d_{f,\mathcal{F}}^{(\rho)}\big(\hat{D}_{\mathcal{S}_i}, \hat{D}_{\mathcal{S}_k}\big) - d_{f,\mathcal{F}}^{(\rho)}\big(\mathcal{D}_{\mathcal{S}_i}, \mathcal{D}_{\mathcal{S}_k}\big) \right| \leq \frac{k}{\rho} \mathfrak{R}_{n_i, \mathcal{D}_{\mathcal{S}_i}}\big(\Pi_{\mathcal{H}}\mathcal{F}\big) + \frac{k}{\rho} \mathfrak{R}_{n_k, \mathcal{D}_{\mathcal{S}_k}}\big(\Pi_{\mathcal{H}}\mathcal{F}\big) + \sqrt{\frac{\log \frac{2}{\delta}}{2n_i}} + \sqrt{\frac{\log \frac{2}{\delta}}{2n_k}} \tag{14}$$

where $n_i$ and $n_k$ correspond to the sample size of $\hat{D}_{\mathcal{S}_i}$ and $\hat{D}_{\mathcal{S}_k}$, respectively. The difference between MDD and its empirical version is bounded by the Rademacher complexity as seen in Eqn 14. With these definitions, we derive our generalization bound based on Rademacher complexity and empirical MDD as below.

**Theorem 3** *Given the same setting as Corollary 1 and Lemma 3, for any $\delta > 0$, with probability $1 - 3\delta$, we obtain the following generalization bound for all $f$:*

$$err_{\mathcal{D}_{U_m}}(f) \leq \sum_{i=1}^{N_s} \pi_i \Big( err_{\hat{D}_{\mathcal{S}_i}}^{\rho}(f) \Big) + d_{f,\mathcal{F}}^{(\rho)}\big(\hat{D}_{\mathcal{S}_{i'}}, \hat{D}_{\mathcal{S}_{k'}}\big) + \gamma + \bar{\lambda} + \frac{k}{\rho} \mathfrak{R}_{n_i, \mathcal{D}_{\mathcal{S}_i}}\big(\Pi_{\mathcal{H}}\mathcal{F}\big)$$

$$+ \frac{k}{\rho} \mathfrak{R}_{n_k, \mathcal{D}_{\mathcal{S}_k}}\big(\Pi_{\mathcal{H}}\mathcal{F}\big) + \sqrt{\frac{\log \frac{2}{\delta}}{2n_{i'}}} + \sqrt{\frac{\log \frac{2}{\delta}}{2n_{k'}}} + \sum_{i=1}^{N_s} \pi_i \left( \frac{2k^2}{\rho} \mathfrak{R}_{n_i, \mathcal{D}_{\mathbf{S}_i}}\big(\Pi_1 \mathcal{F}\big) + \sqrt{\frac{\log \frac{2}{\delta}}{2n_i}} \right) \tag{15}$$

where $\Pi_1 \mathcal{F} \triangleq \big\{ x \mapsto f(x, y) \big| y \in \mathcal{Y}, f \in \mathcal{F} \big\}$ as defined in [51] and 1 (in the subscript) represents constant functions mapping all points to the same class (see Appendix for more details). Theorem 3 thus establishes a relationship between the margin value $\rho$ and the generalization error. The first two terms on the right-hand side exhibit minimal variation with an increase in the margin $\rho$, especially when $\rho$ is small and the hypothesis space is rich, thus reducing the overall right-hand side. However, if $\rho$ exceeds a certain threshold (resulting in a weak classifier), then the first two terms significantly increase, resulting in increase of the overall bound. Choosing a better $\rho$ value thus allows one to obtain a desired error bound, making it more informative than a 0-1 loss-based bound. Inspired by this theoretical framework, we now propose our margin-based adversarial learning algorithm for DG in the next section.

## 5 MADG: Methodology

This section details the methodology of the proposed **MADG** algorithm, a novel adversarial DG algorithm motivated by the theory in Section 4. As seen in Theorem 3, the unseen target domain error is upper-bounded by the convex sum of the empirical source domain errors, $\hat{\epsilon}$ term defined in Eqn. (16), $\gamma$, $\bar{\lambda}$ and complexity terms. We hence aim to minimize the two important terms, viz., the convex sum of the empirical source domain errors and the $\hat{\epsilon}$ term (given below).

$$\hat{\epsilon} = d_{f,\mathcal{F}}^{(\rho)}\big(\hat{D}_{\mathcal{S}_{i'}}, \hat{D}_{\mathcal{S}_{k'}}\big), \ \textbf{s.t.} \ d_{f,\mathcal{F}}^{(\rho)}\big(\hat{D}_{\mathcal{S}_{i'}}, \hat{D}_{\mathcal{S}_{k'}}\big) \geq d_{f,\mathcal{F}}^{(\rho)}\big(\hat{D}_{\mathcal{S}_i}, \hat{D}_{\mathcal{S}_k}\big) \ \forall i, k, i', k' \in \{1, \ldots, N_s\} \tag{16}$$

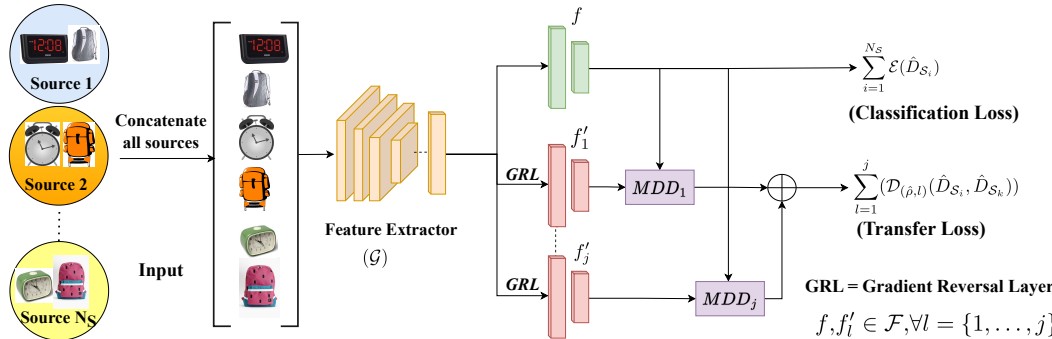

Figure 2: Architecture of the proposed MADG methodology

**Efficient estimation of $\hat{\epsilon}$.** As stated in Eqn. (16), $\hat{\epsilon}$ is defined as the maximum pairwise empirical MDD among all source domains. Similar to Remark 4 and Lemma 2, we can also interpret $\hat{\epsilon}$ as the upper bound of the empirical MDD between any two domains inside the convex hull formed by the source domains. If we minimize the sum of the empirical MDD between different pairs of the source domains, then the size of the convex hull reduces, which in turn minimizes $\hat{\epsilon}$. Thus, one effective way to estimate $\hat{\epsilon}$ is to consider the term below in Eqn. (17).

$$\sum_{i=1}^{N_s-1} \sum_{k=i+1}^{N_s} \mathrm{d}_{f,\mathcal{F}}^{(\rho)}\big(\hat{D}_{S_i}, \hat{D}_{S_k}\big) \tag{17}$$

**Minimax optimization.** To find the optimal $f$ in the hypothesis space $\mathcal{F}$, we formulate our objective function as a minimization problem. As mentioned earlier, we minimize the sum of the empirical source errors and the $\hat{\epsilon}$ term estimated using Eqn (17). Thus, the minimization problem can be written as:

$$\min_{f\in\mathcal{F}} \sum_{i=1}^{N_s} \pi_i\Big(err_{\hat{D}_{S_i}}^{(\rho)}(f)\Big) + \sum_{i=1}^{N_s-1} \sum_{k=i+1}^{N_s} \mathrm{d}_{f,\mathcal{F}}^{(\rho)}\big(\hat{D}_{S_i}, \hat{D}_{S_k}\big) \tag{18}$$

Since empirical MDD is defined as the supremum of the hypothesis space $\mathcal{F}$, minimizing it is in turn a minimax game with a strong max player and weaker min player. In order to strengthen the min player, we use a feature extractor, $\mathcal{G}$, which further modifies the optimization problem as follows:

$$\min_{f,\mathcal{G}} \sum_{i=1}^{N_s} \pi_i\Big(err_{\mathcal{G}(\hat{D}_{S_i})}^{(\rho)}(f)\Big) + \sum_{l=1}^{j} \Big(\mathrm{disp}_{\mathcal{G}(\hat{D}_{S_i})}^{(\rho)}\big(f_l^*, f\big) - \mathrm{disp}_{\mathcal{G}(\hat{D}_{S_k})}^{(\rho)}\big(f_l^*, f\big)\Big),$$
$$f_l^* = \max_{f_l'}\Big(\mathrm{disp}_{\mathcal{G}(\hat{D}_{S_i})}^{(\rho)}\big(f_l', f\big) - \mathrm{disp}_{\mathcal{G}(\hat{D}_{S_k})}^{(\rho)}\big(f_l', f\big)\Big), \text{ where } f, f_l' \in \mathcal{F} \text{ and } \forall l = \{1,\ldots,j\}. \tag{19}$$

where $j = \binom{N_s}{2}$. The relationship between $l, i$ and $k$ is as follows: if $l = 3$ then we pick the $i$ and $k$ value corresponding to the 3rd element of the set $\Big\{(i,k): i = \{1,\ldots,N_s-1\}, k = \{i+1,\ldots,N_s\}\Big\}$. To solve the minimization problem in Eqn. (19), we design an adversarial learning algorithm whose model architecture is shown in Fig. 2. One classifier $f$ performs the classification task on all source domains, while $j$ other classifiers, denoted by $f'$, compute the empirical MDD between different pairs of source domains, which finally sum together as the *transfer loss*. To compute the source errors in Eqn. (19), we use the standard cross-entropy loss $\mathcal{E}(\hat{D}_{S_i})$. For convenience of optimization (MDD can be hard to optimize directly using stochastic gradient descent), in practice, following [40], we approximate the MDD loss as $\Big(\mathcal{D}_{(\hat{\rho},l)}\big(\hat{D}_{S_i}, \hat{D}_{S_k}\big)\Big)$ in terms of two loss functions, $L$ and $L'$, as shown below.

$$\mathcal{E}\big(\hat{D}_{S_i}\big) = \mathbb{E}_{(x,y)\sim\hat{D}_{S_i}}\Big[L\Big(f\big(\mathcal{G}(x)\big), y\Big)\Big]$$

$$\mathcal{D}_{(\hat{\rho},l)}\big(\hat{D}_{S_i}, \hat{D}_{S_k}\big) = \mathbb{E}_{(x,y)\sim\hat{D}_{S_k}}\Big[L'\Big(f_l'\big(\mathcal{G}(x)\big), f\big(\mathcal{G}(x)\big)\Big)\Big] - \hat{\rho}\mathbb{E}_{(x,y)\sim\hat{D}_{S_i}}\Big[L\Big(f_l'\big(\mathcal{G}(x)\big), f\big(\mathcal{G}(x)\big)\Big)\Big] \tag{20}$$

We train the feature extractor, $\mathcal{G}$, to minimize the above MDD loss term by using a Gradient Reversal Layer (GRL) proposed in [52], as $\mathcal{D}_{(\hat{\rho},l)}(\hat{D}_{S_i}, \hat{D}_{S_k})$ is not differentiable w.r.t the parameters of $f$. $L$ and $L'$ are defined as:

$$L\Big(f\big(\mathcal{G}(x)\big),y\Big) \triangleq -\log\Big[\sigma_y\Big(f\big(\mathcal{G}(x)\big)\Big)\Big], \quad L'\Big(f'\big(\mathcal{G}(x)\big),f\big(\mathcal{G}(x)\big)\Big) \triangleq \log\Big[1-\sigma_{h_f\big(\mathcal{G}(x)\big)}\Big(f'\big(\mathcal{G}(x)\big)\Big)\Big]$$
(21)

where $\sigma_w(z) = \frac{e^{z_w}}{\sum_{i=1}^k e^{z_i}}$, $z \in \mathbb{R}^k$, for $w = 1, \ldots, k$, and $\hat{\rho} = e^\rho$. The final optimization problem that the **MADG** method solves is shown below.

$$\min_{f,\mathcal{G}} \sum_{i=1}^{N_s} \pi_i\Big(\mathcal{E}(\hat{D}_{\mathcal{S}_i})\Big) + \sum_{l=1}^{j}\Big(\mathcal{D}_{(\hat{\rho},l)}\big(\hat{D}_{\mathcal{S}_i},\hat{D}_{\mathcal{S}_k}\big)\Big), \quad \max_{f'_1,\ldots,f'_j} \sum_{l=1}^{j}\Big(\mathcal{D}_{(\hat{\rho},l)}\big(\hat{D}_{\mathcal{S}_i},\hat{D}_{\mathcal{S}_k}\big)\Big)$$
(22)

We propose an adversarial DG algorithm to solve the minimization problem, as shown in Algorithm 1. We train the proposed adversarial model in two steps. In the first step, we update the parameters $f$ and $\mathcal{G}$. In the next step, we first do a forward pass to compute the outputs from $f$ and then update the parameters $f'$ and $\mathcal{G}$ as shown in the algorithm. We further show through our ablation studies that this way of updating parameters at different steps outperforms a joint update strategy.

---

**Algorithm 1** Margin-based adversarial learning for Domain Generalization (MADG)

---

INPUT: $N_s$ labeled source domains
**for** epoch $\leftarrow$ 1 to total_epochs **do**
    **for** batch $\leftarrow$ 1 to total_batches **do**
        Update parameters of $f$, $\mathcal{G}$, using Eqn. (20)
        $\hat{y}_{\hat{D}_{\mathcal{S}_i}} \leftarrow f(\mathcal{G}(x)) \; \forall i \in \{1,2,\ldots,N_s\}$
        Compute $MDD_l$ as in Eqn. (20) using $\hat{y}_{\hat{D}_{\mathcal{S}_i}}$
        Compute Transfer loss
            $= \sum_{l=1}^{j}(\mathcal{D}_{(\hat{\rho},l)}\big(\hat{D}_{\mathcal{S}_i},\hat{D}_{\mathcal{S}_k}\big))$
        Update parameters of $f'$ and $\mathcal{G}$ as in Eqn. (22)
    **end for**
**end for**

---

## 6 Experiments

**Datasets and Implementation Details.** We perform an extensive evaluation on five benchmark DG datasets for image classification: VLCS [30], PACS [31], OfficeHome (OH) [32], TerraIncognita (TI) [33] and DomainNet (DN) [34]. We follow [53] in using 'Test-domain validation' procedure for hyperparameter selection. We use the Resnet50 architecture [54] pre-trained on ImageNet dataset [55] as the feature extractor ($\mathcal{G}$). We train our model using stochastic gradient descent with momentum [56]. We use a mini-batch that contains 32 samples from all source domains. All other implementation details including hyperparameters such as learning rate, margin, and weight decay are provided in the Appendix.

**Baselines and evaluation metrics.** We report the results of all baseline models from [35], which are represented by † in Table 1. We also present the results from [53], which are represented by $\perp$. We run our experiments for three trials and report the average accuracy (with standard deviation). Existing state-of-the-art methods (see [35]) focus on average accuracy across the benchmarks, with each method doing well on a subset of datasets. In order to reward consistent performance across datasets while improving average accuracy, we additionally include a ranking-based metric (from [53]) and go beyond ranking with two new metrics (AD and GD), which more precisely capture the consistent performance of a method across datasets: **(i) Median rank (M) [53]:** This measures the median rank of a model's performance across all datasets and is not skewed to best/worst ranks when compared to the mean rank. **(ii) Arithmetic mean of differences (AD):** This measures the difference of a model's performance w.r.t the highest-performing model's accuracy on a given dataset, and then taking its arithmetic mean across datasets. **(iii) Geometric mean of differences (GD):** This similarly measures the geometric mean of the difference between the highest accuracy and the achieved accuracy of a model across all datasets.

**Results.** We report the results on different DG benchmark datasets on the considered evaluation metrics in Table 1. It is observed that most previous methods are far from the best-performing model's accuracy on at least one of the datasets. This is captured well by the metrics AD and GD, which quantify the mean of the differences from the best accuracy across all the datasets. Thus, we observe higher values in the first two columns of Table 1, where our **MADG** model performs the best showing its consistency. The MADG algorithm outperforms all other models on average accuracy, median rank, AD, and GD, thus demonstrating consistent performance.

The results also show that MADG outperform DANN and CDANN models (which are based on the 0-1 loss discrepancy theory) across most datasets and improves the average accuracy by $\approx 2\%$ with significantly better AD and GD values too – corroborating our margin-based approach to the DG problem. MADG is 1% better on the avg accuracy (and lower AD-GD values) when compared to ERM, which was found to be a strong baseline for DG in [35]. Our model also reports an accuracy improvement of $\approx 3\%$ on the OfficeHome dataset compared to all other models.

Table 1: Accuracy(%) on benchmark DG datasets. Values in **red** and orange are best and second best-performing models in a column, respectively. We report deviation as ±N/A for models that did not report them. OH = OfficeHome dataset, TI = TerraIncognita dataset, DN = DomainNet dataset, M = Median rank, AD = Arithmetic mean of differences, GD = Geometric mean of differences, Avg. = Average accuracy(%).

| Algorithm | | AD(↓) | GD(↓) | M(↓) | VLCS | OH | PACS | TI | DN | Avg.(↑) |
|---|---|---|---|---|---|---|---|---|---|---|
| ERM (1998)† [57] | | 2.2 | 1.8 | 8 | 77.6±0.3 | 66.4±0.5 | 86.7±0.3 | 53.0±0.3 | 41.3±0.1 | 65.0 |
| CORAL (2016)† [58] | | 1.9 | 1.1 | **3** | 77.7±0.2 | 68.4±0.2 | 87.1±0.5 | 52.8±0.2 | 41.8±0.1 | 65.6 |
| DANN (2016)† [59] | | 3.4 | 2.1 | 16 | 79.7±0.5 | 65.3±0.8 | 85.2±0.2 | 50.6±0.4 | 38.3±0.1 | 63.8 |
| CDANN (2018)† [60] | | 3.2 | 0.7 | 14 | **79.9±0.2** | 65.3±0.5 | 85.8±0.8 | 50.8±0.6 | 38.5±0.2 | 64.1 |
| MLDG (2018)† [61] | | 2.3 | 1.8 | 6 | 77.5±0.1 | 66.6±0.3 | 86.8±0.4 | 52.0±0.1 | 41.6±0.1 | 64.9 |
| MMD (2018)† [62] | | 5.9 | 0.9 | 10 | 77.9±0.1 | 66.2±0.3 | **87.2±0.1** | 52.0±0.4 | 23.5±9.4 | 61.4 |
| IRM (2019)† [63] | | 6.7 | 5.3 | 18 | 76.9±0.6 | 63.0±2.7 | 84.5±1.1 | 50.5±0.7 | 28.0±5.1 | 60.6 |
| GroupDRO (2019)† [64] | | 3.9 | 1.9 | 10 | 77.4±0.5 | 66.2±0.6 | 87.1±0.1 | 52.4±0.1 | 33.4±0.3 | 63.3 |
| Mixup (2020)† [65] | | 1.9 | 0.4 | 5 | 78.1±0.3 | 68.0±0.2 | 86.8±0.3 | **54.4±0.3** | 39.6±0.1 | 65.4 |
| ARM (2020)† [66] | | 4.1 | 3.4 | 14 | 77.8±0.3 | 64.8±0.4 | 85.8±0.2 | 51.2±0.5 | 36.0±0.2 | 63.1 |
| RSC (2020)† [67] | | 2.9 | 2.5 | 9 | 77.8±0.6 | 66.5±0.6 | 86.2±0.5 | 52.1±0.2 | 38.9±0.6 | 64.3 |
| SagNet (2021)† [68] | | 2.3 | 2.0 | 6 | 77.6±0.1 | 67.5±0.2 | 86.4±0.4 | 52.5±0.4 | 40.8±0.2 | 65.0 |
| V-REx (2021)† [69] | | 4.7 | 0.8 | 12 | 78.1±0.2 | 65.7±0.3 | **87.2±0.6** | 51.4±0.5 | 30.1±3.7 | 62.5 |
| AND-mask (2021)⊥ [70] | | 3.9 | 3.3 | 13 | 76.4±0.4 | 66.1±0.2 | 86.4±0.4 | 49.8±0.4 | 37.9±0.6 | 63.3 |
| Fish (2021) ⊥ [71] | | 2.5 | 0.6 | 13 | 77.8±0.6 | 66.0±2.9 | 85.8±0.6 | 50.8±0.4 | **43.4±0.3** | 64.8 |
| SAND-mask (2021)⊥ [72] | | 5.2 | 4.2 | 16 | 76.2±0.5 | 65.9±0.5 | 85.9±0.4 | 50.2±0.1 | 32.2±0.6 | 62.1 |
| Fishr (2022)⊥ [53] | | 1.5 | 1.2 | **3** | 78.2±0.2 | 68.2±0.2 | 86.9±0.2 | 53.6±0.4 | 41.8±0.2 | 65.7 |
| G2DM (2019)†* [48] | Theory Based Methods | - | - | - | 75.9±N/A | - | 73.6±N/A | - | - | - |
| MTL(2021)†* [47] | | 2.5 | 2.0 | 8 | 77.7±0.5 | 66.5±0.4 | 86.7±0.2 | 52.2±0.4 | 40.8±0.1 | 64.8 |
| Transfer(2021)* [73] | | - | - | - | - | 64.3±N/A | 85.3±N/A | - | - | - |
| Ood(2021)* [49] | | - | - | - | 76.3±N/A | 68.1±N/A | 86.6±N/A | - | - | - |
| **MADG (ours)** | | **1.2** | **0.3** | **3** | 78.7±0.2 | **71.3±0.3** | 86.5±0.4 | 53.7±0.5 | 39.9±0.4 | **66.0** |

To highlight our contribution, we also provide a grouping of DG methods that are theory-based in Table 1 (also denoted with a $*$), some of which do not report results on all benchmark datasets. For the only such method that reports results on all datasets – MTL, MADG achieves 1.2% more average accuracy, 1.3 units less AD, 1.7 units less GD, and a lower median ranking as compared to MTL. More results, including 'Training-domain' model selection results, are provided in the Appendix.

## 7 More Empirical Analysis and Ablation Studies

**Computational Cost.** The computational cost of a model is usually measured in terms of GPU RAM occupied (GB) and time taken per step (s). In terms of computational cost, following Eqn. (17), while we iteratively compute MDD over all domains, our computational cost during training is similar to other benchmark methods as shown in Table 2. As seen in the table, even simple methods like ERM and Mixup have running times in similar ranges.

Table 2: Computational cost and Average Accuracy (%) (Avg.) for different baselines.

| Model | GPU(GB)(↓) | Time(s)(↓) | Avg.(↑) |
|---|---|---|---|
| MADG | ∼10.5 | ∼1.3 | 66.0 |
| Fishr [53] | ∼15.7 | ∼0.6 | 65.7 |
| Fish [71] | ∼3.4 | ∼ 1.2 | 64.8 |
| Mixup [65] | ∼8.2 | ∼0.4 | 65.4 |
| ERM [57] | ∼8.2 | ∼0.4 | 65.0 |

**Margin.** As shown in Theorem 3, the margin value $\rho$ plays an important role, especially with its dependency on the generalization error. In practice, to get the desired margin while training the MADG algorithm, we approximate it as $\hat{\rho} \triangleq \exp \rho$. We experiment with different margin values, $\hat{\rho} = 1$, 1.5, 2, and 3, and report the average accuracy for PACS and VLCS dataset in Table 3. Table 3 shows that $\hat{\rho} = 1.5$ outperforms other values on both datasets. As seen from the definition of the margin loss function Eqn. (1) and MDD Eqn. (5), although having a large margin is better to achieve low loss and higher target accuracy, there is a tradeoff between optimal margin and loss value. We observe a similar trend in Table 3.

**Experiments on Colored MNIST.** As stated in Theorem 2, $\gamma$ is defined as the projection of the unseen domain onto the convex hull. It is also defined in Theorem 2 as the MDD between the convex hull and the unseen domain, which can be further approximated with a combination of two different cross-entropy functions as in Eq (20). This equation is equivalent to a balanced Jensen-Shannon (JS) Divergence as shown in Proposition D.1. in [40]. Thus, we can approximate this projection by computing the JS divergence between domains. We experimented on the Colored MNIST dataset, where we noted the $\gamma$ values to be small for all domains, as shown

Table 3: Accuracy(%) of MADG model for different $\hat{\rho}$ values.

| $\hat{\rho}$ | PACS | VLCS |
|---|---|---|
| 1 | 86.2 | 78.4 |
| **1.5** | **86.5** | **78.7** |
| 2 | 86.2 | 78.4 |
| 3 | 85.7 | 77.8 |

in Table 4. This is because the distributions of different domains in Colored MNIST are relatively close to each other when compared to real-world datasets.

As seen in Table 5, the proposed MADG algorithm achieves 65.6% average accuracy, which is significantly higher than the ERM model (57.8% average accuracy), thus showing that using a margin-based DG algorithm, MADG, learns better domain-invariant features on the Colored MNIST dataset. Besides, Table 4 shows that $\gamma$ is small across domains in this dataset, showcasing the promise of the proposed method when the unseen domain is within the convex hull of the source domains.

Table 4: Pairwise JS divergence for the ColoredMNIST dataset and the approximated $\gamma$ value.

|  | +90 | +80 | -90 | $\sim \gamma$ |
|---|---|---|---|---|
| +90 | 0 | 0.019 | 0.024 | 0.043 |
| +80 | 0.019 | 0 | 0.029 | 0.048 |
| -90 | 0.024 | 0.029 | 0 | 0.054 |

**MDD classifiers.** The proposed MADG algorithm uses 'j' $f'$ classifiers, and each one calculates the MDD between a pair of source domains. In the algorithm, 'j' is defined as $j = \binom{N_S}{2}$. In this section, we analyze the algorithm with $j = N_S - 1$ where $l$ takes corresponding $i$ and $k$ values from $\{(1, i) : i = \{2, \ldots, N_s\}\}$. As seen in Table 6, for '$N_S - 1$', the model performs low across all the datasets, significantly ($\sim 2.5\%$) for the TerraIncognita dataset because the model fails to capture all possible domain discrepancies, leading to poor performance by the feature extractor in learning invariant features.

Table 5: Accuracy (%) on ColoredM-NIST dataset (C).

| C | ERM | MADG |
|---|---|---|
| +90 | 71.8±0.4 | **72.3±0.3** |
| +80 | 72.9±0.1 | **74.0±0.2** |
| -90 | 28.7±0.5 | **50.5±0.4** |
| Average | 57.8 | **65.6** |

**Additional Results.** The proposed MADG algorithm is further compared with recent methods, MIRO [29] and SD [74], which report results using different hyperparameter selection procedures. For a fair comparison, we run both these methods following our setting and report the results on OfficeHome data as shown in Table 7. As seen from Table 7, the proposed MADG outperforms the SD model on all domains. The MADG model also outperforms MIRO on two domains and achieves the same accuracy on the 'R'domain.

Table 6: Accuracy(%) of the MADG model for different MDD classifiers. New = $N_S - 1$, Org. = $\binom{N_S}{2}$

| Data | j=Org. | j=New |
|---|---|---|
| VLCS | **78.7** | 78.5 |
| PACS | **86.5** | 86.4 |
| OH | **71.3** | 70.8 |
| TI | **53.7** | 51.1 |

**Weighted MDD.** The proposed MADG algorithm calculates the Transfer loss as the arithmetic sum of the 'j' MDD losses. This section analyzes the effect on the model's performance using a weighted ($w$) arithmetic sum. The different weights used in this section are $w_l = \frac{1}{N_S}, \forall l = 1 \ldots j$(Average) and $w_l = \frac{MDD_L}{\sum_{l=1}^{j} MDD_l}$ (Dynamic). The performance of the MADG algorithm using different weights is reported in Table 8. The model with $w_l = 1, \forall l = 1 \ldots j$ performs better than other weight values as reported in Table 8.

Table 7: Accuracy(%) on OfficeHome (OH) dataset

| OH | Miro | SD | MADG |
|---|---|---|---|
| A | 67.0 ±0.7 | 64.8±0.9 | **68.6 ±0.5** |
| C | **56.5 ±0.9** | 49.9 ±1.2 | 55.5 ±0.2 |
| P | 79.4 ±1.4 | 75.6 ±1.3 | **79.6 ±0.3** |
| R | **81.5 ±0.5** | 79.1 ±0.2 | **81.5 ±0.4** |
| Avg | 71.1 | 67.4 | **71.3** |

# 8 Conclusion

This study presented a novel adversarial algorithm for DG called **MADG**. Our approach is inspired by a theoretical framework that formulates the relationship between the source and unseen domains in real-valued hypothesis space. We used the margin loss function and Rademacher complexity framework to develop a generalization bound, which is informative. The proposed bound also shows a dependency between the margin value and the generalization error, which can be easily translated into an algorithm. To this end, we developed the **MADG** algorithm that leverages the MDD metric with minimax optimization to learn domain-invariant features and generalize well to the unseen domain. The algorithm was studied on five well-known DG benchmark datasets and reported consistent performance across all datasets as compared to other methods. We also report several empirical analyses and ablation studies.

Table 8: Accuracy(%) of the MADG with different $w_l$ values. One: $w_l = 1$, Avg.=Average, Dyn.=Dynamic

|  | One | Avg. | Dyn. |
|---|---|---|---|
| P | 97.7 | **97.8** | 97.6 |
| A | **87.8** | 87.3 | 87.5 |
| C | 82.2 | 82.0 | **82.4** |
| S | **78.3** | 77.4 | 77.2 |
| Avg | **86.5** | 86.1 | 86.2 |

## Acknowledgments

The research of AD is supported in part by the Prime Minister Research Fellowship, Ministry of Education, Government of India. The research of LRC is supported by the Indo-Norwegian Collaboration in Autonomous Cyber-Physical Systems (INCAPS) project: 287918 of the International Partnerships for Excellent Education, Research and Innovation (INTPART) program and the Low-Altitude UAV Communication and Tracking (LUCAT) project: 280835 of the IKTPLUSS program from the Research Council of Norway. The research of AK is supported by TiHAN Faculty Fellowship. VNB would like to acknowledge the support through the Govt of India SERB IMPRINT and DST ICPS funding programs for this work. We are grateful to the anonymous reviewers for the feedback that helped improved the presentation of this paper.

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
