# APPENDIX
# MADG: Margin-based Adversarial Learning for Domain Generalization

In this appendix, we aim to provide a more comprehensive understanding of our contributions by presenting information, such as proofs, implementation details, additional results, and analysis that we could not include in the main paper due to space constraints. Below is a specific list of information included in this appendix.

## A1 Proofs

**Lemma 1.** *Consider a weighted sum of $N_s - 1$ labeled source distributions defined as $\mathcal{D}_{\bar{\mathcal{S}}} := \sum_{i=1}^{N_s-1} \alpha_i \mathcal{D}_{\mathcal{S}_i}$, where $\alpha_i$s are mixture co-efficients s.t. $\sum_{i=1}^{N_s-1} \alpha_i = 1$, and $f$ is a scoring function. Then*

$$d_{f,\mathcal{F}}^{(\rho)}\big(\mathcal{D}_{\bar{\mathcal{S}}}, \mathcal{D}_{\mathcal{S}_T}\big) \leq \sum_{i=1}^{N_s-1} \alpha_i d_{f,\mathcal{F}}^{(\rho)}\big(\mathcal{D}_{\mathcal{S}_i}, \mathcal{D}_{\mathcal{S}_T}\big)$$

Proof: Consider a mixture distribution of $N_s - 1$ source domains with support $\Omega = \mathcal{X} \times \mathcal{Y}$. Then

$$\mathrm{d}_{f,\mathcal{F}}^{(\rho)}(\mathcal{D}_{\bar{\mathcal{S}}}, \mathcal{D}_{\mathcal{S}_T}) = \sup_{f' \in \mathcal{F}} \left( \mathbb{E}_{\mathcal{D}_{\mathcal{S}_T}}\Big[ \Phi_\rho \circ \rho_{f'}(.,h_f) \Big] - \mathbb{E}_{\mathcal{D}_{\bar{\mathcal{S}}}}\Big[ \Phi_\rho \circ \rho_{f'}(.,h_f) \Big] \right)$$

$$= \sup_{f' \in \mathcal{F}} \left( \iint_\Omega \mathcal{D}_{\mathcal{S}_T}(.) \, \Phi_\rho \circ \rho_{f'}(x,h_f) \, d\Omega - \iint_\Omega \mathcal{D}_{\bar{\mathcal{S}}}(.) \, \Phi_\rho \circ \rho_{f'}(x,h_f) d\Omega \right)$$

$$= \sup_{f' \in \mathcal{F}} \left( \iint_\Omega \mathcal{D}_{\mathcal{S}_T}(.) \, \Phi_\rho \circ \rho_{f'}(x,h_f) \, d\Omega - \iint_\Omega \sum_{i=1}^{N_s-1} \alpha_i \mathcal{D}_{\mathcal{S}_i}(.) \, \Phi_\rho \circ \rho_{f'}(x,h_f) \, d\Omega \right)$$

$$\text{since } \sum_{i=1}^{N_s-1} \alpha_i = 1$$

$$= \sup_{f' \in \mathcal{F}} \left( \iint_\Omega \sum_{i=1}^{N_s-1} \alpha_i \mathcal{D}_{\mathcal{S}_T}(.) \, \Phi_\rho \circ \rho_{f'}(x, h_f) \, d\Omega - \iint_\Omega \sum_{i=1}^{N_s-1} \alpha_i \mathcal{D}_{\mathcal{S}_i}(.) \, \Phi_\rho \circ \rho_{f'}(x, h_f) \, d\Omega \right)$$

$$= \sup_{f' \in \mathcal{F}} \sum_{i=1}^{N_s-1} \alpha_i \left( \iint_\Omega \mathcal{D}_{\mathcal{S}_T}(.) \, \Phi_\rho \circ \rho_{f'}(x, h_f) \, d\Omega - \iint_\Omega \mathcal{D}_{\mathcal{S}_i}(.) \, \Phi_\rho \circ \rho_{f'}(x, h_f) \, d\Omega \right)$$

From sub-additivity of sup function, we get:

$$\leq \alpha_i \sum_{i=1}^{N_s-1} \sup_{f' \in \mathcal{F}} \left( \iint_\Omega \mathcal{D}_{\mathcal{S}_T}(.) \, \Phi_\rho \circ \rho_{f'}(x, h_f) \, d\Omega - \iint_\Omega \mathcal{D}_{\mathcal{S}_i}(.) \, \Phi_\rho \circ \rho_{f'}(x, h_f) \, d\Omega \right)$$

$$= \sum_{i=1}^{N_s-1} \alpha_i d_{f,\mathcal{F}}^{(\rho)}(\mathcal{D}_{\mathcal{S}_i}, \mathcal{D}_{\mathcal{S}_T})$$

Thus, we have:

$$d_{f,\mathcal{F}}^{(\rho)}(\mathcal{D}_{\bar{\mathcal{S}}}, \mathcal{D}_{\mathcal{S}_T}) \leq \sum_{i=1}^{N_s-1} \alpha_i d_{f,\mathcal{F}}^{(\rho)}(\mathcal{D}_{\mathcal{S}_i}, \mathcal{D}_{\mathcal{S}_T}) \qquad\qquad \square$$

**Theorem 1.** *Consider a scoring function $f$, unlabeled source domain $\mathcal{D}_{\mathcal{S}_T}$ and a mixture of $N_s - 1$ source distributions denoted as $\mathcal{D}_{\bar{\mathcal{S}}} := \sum_{i=1}^{N_s-1} \alpha_i \mathcal{D}_{\mathcal{S}_i}$, where $\alpha_i s$ is mixture co-efficients s.t. $\sum_{i=1}^{N_s-1} \alpha_i = 1$. Then the error on the unlabeled source $\mathcal{D}_{\mathcal{S}_T}$ is bounded as:*

$$err_{\mathcal{D}_{\mathcal{S}_T}}(h_f) \leq \sum_{i=1}^{N_s-1} \alpha_i \left( err_{\mathcal{D}_{\mathcal{S}_i}}^{(\rho)}(f) + d_{f,\mathcal{F}}^{(\rho)}(\mathcal{D}_{\mathcal{S}_i}, \mathcal{D}_{\mathcal{S}_T}) \right) + \hat{\lambda}$$

Proof: For a setting with a single labeled and unlabeled domain in the DA setting, it has been shown that the error bound on the unlabeled domain is bounded as follows [40]:

$$err_{\mathcal{D}_{\mathcal{S}_T}}(h_f) \leq \left( err_{\mathcal{D}_S}^{(\rho)}(f) + d_{f,\mathcal{F}}^{(\rho)}(\mathcal{D}_S, \mathcal{D}_{\mathcal{S}_T}) \right) + \lambda$$

where $\lambda$ is the ideal combined margin loss independent of the scoring function and is defined as $\lambda = \min_{f^* \in \mathcal{F}} \left( err_{\mathcal{D}_S}^{(\rho)}(f^*) + err_{\mathcal{D}_{\mathcal{S}_T}}^{(\rho)}(f^*) \right)$. We herein leverage this to provide an upper bound for the error on an unlabeled source domain when multiple labeled source domains are available, as shown below.

$$err_{\mathcal{D}_{\mathcal{S}_T}}(h_f) \leq (err_{\mathcal{D}_{\bar{\mathcal{S}}}}^{(\rho)}(f) + d_{f,\mathcal{F}}^{(\rho)}(\mathcal{D}_{\bar{\mathcal{S}}}, \mathcal{D}_{\mathcal{T}}) + \hat{\lambda}$$

where $\hat{\lambda} = err_{\mathcal{D}_{\bar{\mathcal{S}}}}^{(\rho)}(f^*) + err_{\mathcal{D}_{\mathcal{S}_T}}^{(\rho)}(f^*))$. We know that $\forall f \in \mathcal{F}$:

$$err_{\mathcal{D}_{\bar{\mathcal{S}}}}^{(\rho)}(f) = \sum_{i=1}^{N_s-1} \alpha_i err_{\mathcal{D}_{\mathcal{S}_i}}^{(\rho)}(f)$$

From the above result, we get:

$$err_{\mathcal{D}_{\mathcal{S}_T}}(h_f) \leq \left( \sum_{i=1}^{N_s-1} \left( \alpha_i err_{\mathcal{D}_{\mathcal{S}_i}}^{(\rho)}(f) \right) + d_{f,\mathcal{F}}^{(\rho)}(\mathcal{D}_{\bar{\mathcal{S}}}, \mathcal{D}_{\mathcal{S}_T}) \right) + \hat{\lambda}$$

From Lemma 1, we get an upper bound on the MDD between the mixture distribution and unlabeled source domain. Therefore the upper bound for the unlabeled source error is given by:

$$err_{\mathcal{D}_{\mathcal{S}_T}}(h_f) \leq \sum_{i=1}^{N_s-1} \alpha_i \left( err_{\mathcal{D}_{\mathcal{S}_i}}^{(\rho)}(f) + d_{f,\mathcal{F}}^{(\rho)}(\mathcal{D}_{\mathcal{S}_i}, \mathcal{D}_{\mathcal{S}_T}) \right) + \hat{\lambda} \qquad\qquad \square$$

**Lemma 2.** *Let $d_{f,\mathcal{F}}^{(\rho)}(\mathcal{D}_{\mathcal{S}_i}, \mathcal{D}_{\mathcal{S}_k}) \leq \epsilon \; \forall i, k \in \{1, 2, \ldots, N_s\}$ and $f$ be a scoring function. Then the following inequality holds for the MDD between any pair of domains $\mathcal{D}', \mathcal{D}'' \in \Lambda_s$:*

$$d_{f,\mathcal{F}}^{(\rho)}(\mathcal{D}', \mathcal{D}'') \leq \epsilon$$

Proof: Consider two domains, $\mathcal{D}'$ and $\mathcal{D}''$ on the convex hull $\Lambda_S$ of $N_s$ source domains with support $\Omega = \mathcal{X} \times \mathcal{Y}$. Consider also $\mathcal{D}' = \sum_{i=1}^{N_s} \pi_i \mathcal{D}_{\mathcal{S}_i}(.)$ and $\mathcal{D}'' = \sum_{k=1}^{N_s} \pi_k \mathcal{D}_{\mathcal{S}_k}(.)$, where $\sum_{i=1}^{N_s} \pi_i = 1$ and $\sum_{k=1}^{N_s} \pi_k = 1,$. The MDD between $\mathcal{D}'$ and $\mathcal{D}''$ can be written as:

$$d_{f,\mathcal{F}}^{(\rho)}(\mathcal{D}', \mathcal{D}'') = \sup_{f' \in \mathcal{F}} \left( \mathbb{E}_{\mathcal{D}''} \left[ \Phi_\rho \circ \rho_{f'}(., h_f) \right] - \mathbb{E}_{\mathcal{D}'} \left[ \Phi_\rho \circ \rho_{f'}(., h_f) \right] \right)$$

$$= \sup_{f' \in \mathcal{F}} \left( \iint_\Omega \mathcal{D}''(.) \, \Phi_\rho \circ \rho_{f'}(x, h_f) \, d\Omega - \iint_\Omega \mathcal{D}'(.) \, \Phi_\rho \circ \rho_{f'}(x, h_f) \, d\Omega \right)$$

$$= \sup_{f' \in \mathcal{F}} \left( \iint_\Omega \sum_{k=1}^{N_s} \pi_k \mathcal{D}_{\mathcal{S}_k}(.) \, \Phi_\rho \circ \rho_{f'}(x, h_f) \, d\Omega - \iint_\Omega \sum_{i=1}^{N_s} \pi_i \mathcal{D}_{\mathcal{S}_i}(.) \, \Phi_\rho \circ \rho_{fx}(x, h_f) \, d\Omega \right)$$

Since $\sum_{i=1}^{N_s} \pi_i = 1 \text{ and } \sum_{k=1}^{N_s} \pi_k = 1, \text{ we get :}$

$$= \sup_{f' \in \mathcal{F}} \left( \iint_\Omega \sum_{k=1}^{N_s} \sum_{i=1}^{N_s} \pi_k \pi_i \mathcal{D}_{\mathcal{S}_k}(.) \, \Phi_\rho \circ \rho_{f'}(x, h_f) \, d\Omega - \iint_\Omega \sum_{i=1}^{N_s} \sum_{k=1}^{N_s} \pi_i \pi_k \mathcal{D}_{\mathcal{S}_i}(.) \, \Phi_\rho \circ \rho_{f'}(x, h_f) \, d\Omega \right)$$

$$= \sup_{f' \in \mathcal{F}} \sum_{i=1}^{N_s} \sum_{k=1}^{N_s} \pi_i \pi_k \left( \iint_\Omega \mathcal{D}_{\mathcal{S}_k}(.) \, \Phi_\rho \circ \rho_{f'}(x, h_f) \, d\Omega - \iint_\Omega \mathcal{D}_{\mathcal{S}_i}(.) \, \Phi_\rho \circ \rho_{f'}(x, h_f) \, d\Omega \right)$$

From sub-additivity of the sup function:

$$\leq \sum_{i=1}^{N_s} \sum_{k=1}^{N_s} \pi_i \pi_k \sup_{f' \in \mathcal{F}} \left( \iint_\Omega \mathcal{D}_{\mathcal{S}_k}(.) \, \Phi_\rho \circ \rho_{f'}(x, h_f) \, d\Omega - \iint_\Omega \mathcal{D}_{\mathcal{S}_i}(.) \, \Phi_\rho \circ \rho_{f'}(x, h_f) \, d\Omega \right)$$

$$= \sum_{i=1}^{N_s} \sum_{k=1}^{N_s} \pi_i \pi_k \left( d_{f,\mathcal{F}}^{(\rho)}(\mathcal{D}_{\mathcal{S}_i}, \mathcal{D}_{\mathcal{S}_k}) \right)$$

Given that $d_{f,\mathcal{F}}^{(\rho)}(\mathcal{D}_{\mathcal{S}_i}, \mathcal{D}_{\mathcal{S}_k}) \leq \epsilon \; \forall i, k \in \{1, 2, \ldots, N_s\}$, we get:

$$d_{f,\mathcal{F}}^{(\rho)}(\mathcal{D}', \mathcal{D}'') \leq \epsilon \qquad \square$$

**Theorem 2.** *Consider a mixture of $N_s$ source distributions, scoring function f, unseen domain $\mathcal{D}_{U_m}$, and $\gamma = d_{f,\mathcal{F}}^{(\rho)}(\bar{\mathcal{D}}_U, \mathcal{D}_{U_m})$ where $\bar{\mathcal{D}}_U$ is the projection of $\mathcal{D}_{U_m}$ onto the convex hull of the sources i.e. $\bar{\mathcal{D}}_U = argmin_{\pi_1, \pi_2, \ldots, \pi_{N_s}} d_{f,\mathcal{F}}^{(\rho)}\left( \sum_{i=1}^{N_s} \pi_i \mathcal{D}_{\mathcal{S}_i}, \mathcal{D}_{U_m} \right), \sum_{i=1}^{N_s} \pi_i = 1$. Then, the unseen target error is bounded as follows:*

$$err_{\mathcal{D}_{U_m}}(h_f) \leq \sum_{i=1}^{N_s} \pi_i \left( err_{\mathcal{D}_{\mathcal{S}_i}}^{(\rho)}(f) \right) + \epsilon + \gamma + \bar{\lambda}$$

Proof: Motivated by Theorem 1, we now analyze the error for an unseen target domain and provide a bound w.r.t. a mixture distribution of $N_s$ labeled source domains as shown below.

$$err_{\mathcal{D}_{U_m}}(h_f) \leq \sum_{i=1}^{N_s} \pi_i (err_{\mathcal{S}_i}^{\rho}(f) + d_{f,\mathcal{F}}^{(\rho)}(\mathcal{D}_{\mathcal{S}_i}, \mathcal{D}_{U_m})) + \bar{\lambda}$$

Using sub-additivity of the sup function, we get:

$$d_{f,\mathcal{F}}^{(\rho)}(\mathcal{D}_{\mathcal{S}_i}, \mathcal{D}_{U_m}) \leq d_{f,\mathcal{F}}^{(\rho)}(\mathcal{D}_{\mathcal{S}_i}, \bar{\mathcal{D}}_U) + d_{f,\mathcal{F}}^{(\rho)}(\bar{\mathcal{D}}_U, \mathcal{D}_{U_m})$$

Using Lemma 2 and the definition of $\gamma$:

$$d_{f,\mathcal{F}}^{(\rho)}(\mathcal{D}_{\mathcal{S}_i}, \mathcal{D}_{U_m}) \leq \epsilon + \gamma$$

Thus, we now upper-bound $\sum_{i=1}^{N_s} \pi_i d_{f,\mathcal{F}}^{(\rho)}(\mathcal{D}_{\mathcal{S}_i}, \mathcal{D}_{U_m})$ by $\epsilon + \gamma$. Also, in Theorem 1, $\hat{\lambda}$ depended on the joint ideal loss between multiple labeled sources and the unlabeled one. Since we have an unseen target in DG, we define $\bar{\lambda} = \min_{f^* \in \mathcal{F}} \left( \sum_{i=1}^{N_s} \pi_i err_{\mathcal{D}_{\mathcal{S}_i}}^{(\rho)}(f^*) + err_{\mathcal{D}_{U_m}}^{(\rho)}(f^*) \right)$, which is the joint ideal error between multiple sources and unseen domain. Thus, using the above results, we can upper bound the unseen target error as shown in Theorem 2. $\qquad \square$

**Corollary 1.** *Consider a mixture of $N_s$ source distributions, scoring function f, unseen domain $\mathcal{D}_{U_m}$, $\gamma = d_{f,\mathcal{F}}^{(\rho)}(\mathcal{D}_{U_m}, \bar{\mathcal{D}}_U)$, and $d_{f,\mathcal{F}}^{(\rho)}(\mathcal{D}_{S_{i'}}, \mathcal{D}_{S_{k'}})$ where $d_{f,\mathcal{F}}^{(\rho)}(\mathcal{D}_{S_{i'}}, \mathcal{D}_{S_{k'}}) \geq d_{f,\mathcal{F}}^{(\rho)}(\mathcal{D}_{S_i}, \mathcal{D}_{S_k}) \, \forall i, k, i', k' \in \{1, 2, \ldots, N_s\}$. Then the unseen target error is bounded as follows:*

$$err_{\mathcal{D}_{U_m}}(h_f) \leq \sum_{i=1}^{N_s} \pi_i \left( err_{\mathcal{D}_{S_i}}^{(\rho)}(f) \right) + d_{f,\mathcal{F}}^{(\rho)}(\mathcal{D}_{S_{i'}}, \mathcal{D}_{S_{k'}}) + \gamma + \bar{\lambda}$$

**Lemma 3.** *[40]. For any $\delta > 0$, with probability $1 - 2\delta$, the following holds simultaneously for any scoring function $f$:*

$$\left| d_{f,\mathcal{F}}^{(\rho)}(\hat{D}_{S_i}, \hat{D}_{S_k}) - d_{f,\mathcal{F}}^{(\rho)}(\mathcal{D}_{S_i}, \mathcal{D}_{S_k}) \right| \leq \frac{k}{\rho} \mathfrak{R}_{n_i, \mathcal{D}_{S_i}}(\Pi_{\mathcal{H}}\mathcal{F}) + \frac{k}{\rho} \mathfrak{R}_{n_k, \mathcal{D}_{S_k}}(\Pi_{\mathcal{H}}\mathcal{F}) + \sqrt{\frac{\log \frac{2}{\delta}}{2n_i}} + \sqrt{\frac{\log \frac{2}{\delta}}{2n_k}}$$

**Lemma 4.** *[40]. Let $\Pi_1\mathcal{F} = \{x \mapsto f(x, y) | y \in \mathcal{Y}, f \in \mathcal{F}\}$. Then for any $\delta > 0$, for a fixed $\rho > 0$, with probability atleast $1 - \delta$, the following holds for all scoring function $f \in \mathcal{F}$:*

$$|err_{\mathcal{D}_{S_i}}^{\rho}(f) - err_{\hat{\mathcal{D}}_{S_i}}^{\rho}(f)| \leq \frac{2k^2}{\rho} \mathfrak{R}_{n_i, \mathcal{D}_{S_i}}(\Pi_1\mathcal{F}) + \sqrt{\frac{\log \frac{2}{\delta}}{2n_i}}$$

*A simple corollary of this lemma is as follows:*

$$err_{\mathcal{D}_{S_i}}^{\rho}(h_f) \leq err_{\mathcal{D}_{S_i}}^{\rho}(f)$$

$$\leq err_{\hat{\mathcal{D}}_{S_i}}^{\rho}(f) + \frac{2k^2}{\rho} \mathfrak{R}_{n_i, \mathcal{D}_{S_i}}(\Pi_1\mathcal{F}) + \sqrt{\frac{\log \frac{2}{\delta}}{2n_i}}$$

As defined in [40], the term $\Pi_{\mathcal{H}}\mathcal{F}$ can be interpreted as the inner product between the vector fields $\mathcal{H}$ and $\mathcal{F}$ defined as $\Pi_{\mathcal{H}}\mathcal{F} = \langle \mathcal{H}, \mathcal{F} \rangle = \{\langle h, f \rangle | h \in \mathcal{H}, f \in \mathcal{F}\}$. Similarly $\Pi_1\mathcal{F}$ can be understood as the union of projections of $\mathcal{F}$ onto each dimension.

**Theorem 3.** *Given the same setting as Corollary 1 and Lemma 3, for any $\delta > 0$, with probability $1 - 3\delta$, we obtain the following generalization bound for all $f$:*

$$err_{\mathcal{D}_{U_m}}(f) \leq \sum_{i=1}^{N_s} \pi_i \left( err_{\hat{D}_{S_i}}^{\rho}(f) \right) + d_{f,\mathcal{F}}^{(\rho)}(\hat{D}_{S_{i'}}, \hat{D}_{S_{k'}}) + \gamma + \bar{\lambda} + \frac{k}{\rho} \mathfrak{R}_{n_i, \mathcal{D}_{S_i}}(\Pi_{\mathcal{H}}\mathcal{F})$$

$$+ \frac{k}{\rho} \mathfrak{R}_{n_k, \mathcal{D}_{S_k}}(\Pi_{\mathcal{H}}\mathcal{F}) + \sqrt{\frac{\log \frac{2}{\delta}}{2n_{i'}}} + \sqrt{\frac{\log \frac{2}{\delta}}{2n_{k'}}} + \sum_{i=1}^{N_s} \pi_i \left( \frac{2k^2}{\rho} \mathfrak{R}_{n_i, \mathbf{\mathcal{D}}_{S_i}}(\Pi_1\mathcal{F}) + \sqrt{\frac{\log \frac{2}{\delta}}{2n_i}} \right)$$

Proof: From Lemma 4, for $N_s$ source distributions, the expected margin loss is upper bounded, as shown below.

$$\sum_{i=1}^{N_s} \pi_i \left( err_{\mathcal{D}_{S_i}}^{\rho}(f) \right) \leq \sum_{i=1}^{N_s} \pi_i \left( err_{\hat{\mathcal{D}}_{S_i}}^{\rho}(f) + \frac{2k^2}{\rho} \mathfrak{R}_{n_i, \mathcal{D}_{S_i}}(\Pi_1\mathcal{F}) + \sqrt{\frac{\log \frac{2}{\delta}}{2n_i}} \right).$$

From Lemma 3, we upper-bound the expected MDD as shown below.

$$d_{f,\mathcal{F}}^{(\rho)}(\mathcal{D}_{S_i}, \mathcal{D}_{S_k}) \leq d_{f,\mathcal{F}}^{(\rho)}(\hat{D}_{S_i}, \hat{D}_{S_k}) + \frac{k}{\rho} \mathfrak{R}_{n_i, \mathcal{D}_{S_i}}(\Pi_{\mathcal{H}}\mathcal{F}) + \frac{k}{\rho} \mathfrak{R}_{n_k, \mathcal{D}_{S_k}}(\Pi_{\mathcal{H}}\mathcal{F}) + \sqrt{\frac{\log \frac{2}{\delta}}{2n_i}} + \sqrt{\frac{\log \frac{2}{\delta}}{2n_k}}$$

Using the above results and Corollary 1, we get Theorem 3 □.

## A2   Other Related Work

In this section, we discuss earlier literature proposed for the DG problem, in terms of their broad categories. Apart from adversarial learning, existing DG methods can be broadly categorized into: (i) *Meta-Learning* approaches, where the model is exposed to domain shift during training to equip it with better domain adaptation [A1–A6]; (ii) *Data augmentation* techniques that generate multiple augmentations of a data sample to help the model overcome overfitting and achieve better generalization [A7–A12]; (iii) *Self-Supervised learning* methods that involve training the model to predict a pretext task first and then fine-tuning it for the downstream task [A13–A18]; and (iv) *Regularization* techniques that aim to restrict the model with certain constraints that enable it to learn general features and prevent it from learning domain-specific features [A19–A25].

## A3  Implementation Details

This section provides detailed information about the implementation of our experiments on the five benchmark datasets for the DG problem. The datasets and their respective details are outlined as follows, **VLCS:** This consists of 4 domains, namely, VOC2007, LabelMe, Caltech101, SUN09, with a total of 10,729 images, each with dimension (3,224,224) and 5 classes. **OfficeHome:** There are 4 domains in this dataset: Art, Clipart, Product, and Real, with 15,588 examples and 65 classes. Each sample image has a dimension of (3,224,224). **PACS:** It comprises of 4 domains, Photo, Art, Cartoon, and Sketch, with 9,991 samples, each with dimension (3,224,224) and 7 classes. **TerraIncognita:** This dataset focuses on wild animal images taken by camera traps at 4 locations, L100, L43, L46, and L38, with 24,788 examples each with dimensions (3,224,224) and 10 classes. **DomainNet:** There are a total of 6 domains in this dataset, clipart, sketch, infograph, quickdraw, painting, and real, with 586,575 images each with dimension (3,224,224) and 345 classes.

By conducting experiments on these benchmark datasets, we ensure a comprehensive evaluation of the proposed MADG model's performance, taking into account diverse domains, varying class distributions, and image characteristics. The code for our work is available at: https://github.com/qwedaq/MADG.

The MADG model uses ResNet50 [54] architecture pre-trained on the ImageNet [55] dataset as the feature extractor ($\mathcal{G}$). The model is trained using stochastic gradient descent with momentum optimizer [56] on Nvidia V100 GPU with 32GB RAM. We list all hyperparameter details of the MADG model according to the dataset in Table A1. We also list the search space for each hyperparameter as shown in Table A2. We follow the recent state-of-the-art [53] (also supported by [A26–A28]) to tune the hyperparameters by selecting a hyperparameter value that performs the best from a range of values shown in Table A2.

Table A1: Hyperparameter values of our MADG model implementation on all datasets

| Hyperparameter | VLCS | OfficeHome | PACS | TerraIncognita | DomainNet |
|---|---|---|---|---|---|
| Margin | 1.5 | 2 | 1.5 | 2 | 2 |
| Learning rate | 0.004 | 0.004 | 0.004 | 0.01 | 0.004 |
| Momentum | 0.9 | 0.9 | 0.9 | 0.9 | 0.9 |
| Weight decay | 0.0005 | 0.0005 | 0.0005 | 0.005 | 0.0005 |
| Dropout | 0.5 | 0.5 | 0.5 | 0.3 | 0.5 |
| Learning rate decay | 0.0002 | 0.0002 | 0.0002 | 0.001 | 0.0002 |

Table A2: Ranges of values for hyperparameter tuning.

| **Hyperparameter** | **Search Values** |
|---|---|
| Margin | [1, 1.5, 2, 3] |
| Learning rate | [0.01, 0.04, 0.001, 0.004, 0.0001] |
| Momentum | [0.85, 0.9, 0.95] |
| Weight Decay | [0.005, 0.0005] |
| Dropout | [0.3, 0.4, 0.45, 0.5] |
| Learning rate decay | [0.001, 0.0002] |

## A4  Additional Results

This section provides detailed results for each domain on all five datasets in Tables A3 to A7. Note that for the Fish [71] model, no per-dataset results are available. As stated in the main paper, we observe from these detailed results that the proposed MADG provides consistently strong results across the domains and benchmarks, as evidenced by the AD, GD and M metrics in Table 1 (main paper).

Table A3: Accuracy (%) on each domain of the OfficeHome dataset. We represent '-' for models which did not report accuracy for that domain. We also report deviation as ±N/A for models that did not report them.

| Algorithm | A | C | P | R | Avg.(↑) |
|---|---|---|---|---|---|
| ERM (1998)[57] | 61.7±0.7 | 53.4±0.3 | 74.1±0.4 | 76.2±0.6 | 66.4 |
| CORAL (2016)[58] | 64.8±0.8 | 54.1±0.9 | 76.5±0.4 | 78.2±0.4 | 68.4 |
| DANN (2016)[59] | 60.6±1.4 | 51.8±0.7 | 73.4±0.5 | 75.5±0.9 | 65.3 |
| CDANN (2018)[60] | 57.9±0.2 | 52.1±1.2 | 74.9±0.7 | 76.2±0.2 | 65.3 |
| MLDG (2018)[61] | 60.5±0.7 | 54.2±0.5 | 75.0±0.2 | 76.7±0.5 | 66.6 |
| MMD (2018)[62] | 60.4±1.0 | 53.4±0.5 | 74.9±0.1 | 76.1±0.7 | 66.2 |
| IRM (2019)[63] | 56.4±3.2 | 51.2±2.3 | 71.7±2.7 | 72.7±2.7 | 63.0 |
| GroupDRO (2019)[64] | 60.5±1.6 | 53.1±0.3 | 75.5±0.3 | 75.9±0.7 | 66.2 |
| Mixup (2020)[65] | 63.5±0.2 | 54.6±0.4 | 76.0±0.3 | 78.0±0.7 | 68.0 |
| ARM (2020)[66] | 58.8±0.5 | 51.8±0.7 | 74.0±0.1 | 74.4±0.2 | 64.8 |
| RSC (2020)[67] | 61.7±0.8 | 53.0±0.9 | 74.8±0.8 | 76.3±0.5 | 66.5 |
| SagNet (2021)[68] | 62.7±0.5 | 53.6±0.5 | 76.0±0.3 | 77.8±0.1 | 67.5 |
| V-REx (2021)[69] | 59.6±1.0 | 53.3±0.3 | 73.2±0.5 | 76.6±0.4 | 65.7 |
| AND-mask (2021)[70] | 60.3±0.5 | 52.3±0.6 | 75.1±0.2 | 76.6±0.3 | 66.1 |
| Fish (2021) [71] | - | - | - | - | 66.0 |
| SAND-mask (2021)[72] | 59.9±0.7 | 53.6±0.8 | 74.3±0.4 | 75.8±0.5 | 65.9 |
| Fishr (2022)[53] | 63.4±0.8 | 54.2±0.3 | 76.4±0.3 | 78.5±0.2 | 68.2 |
| GDM(2019)[48] | - | - | - | - | - |
| MTL(2021)[47] | 60.7±0.8 | 53.5±1.3 | 75.2±0.6 | 76.6±0.6 | 66.5 |
| Transfer(2021)[73] | - | - | - | - | 64.3 |
| Ood(2021)[49] | 65.8±N/A | 55.1±N/A | 75.2±N/A | 76.3±N/A | 68.1 |
| **MADG (ours)** | 68.6±0.5 | 55.5±0.2 | 79.6±0.3 | 81.5±0.4 | 71.3 |

(Theory-Based Methods label spans the GDM, MTL, Transfer, Ood rows)

Table A4: Accuracy (%) on each domain of the TerraIncognita dataset. We represent '-' for models which did not report accuracy for that domain. We also report deviation as ±N/A for models that did not report them.

| Algorithm | L100 | L38 | L43 | L46 | Avg.(↑) |
|---|---|---|---|---|---|
| ERM (1998)[57] | 59.4±0.9 | 49.3±0.6 | 60.1±1.1 | 43.2±0.5 | 53.0 |
| CORAL (2016)[58] | 60.4±0.9 | 47.2±0.5 | 59.3±0.4 | 44.4±0.4 | 52.8 |
| DANN (2016)[59] | 55.2±1.9 | 47.0±0.7 | 57.2±0.9 | 42.9±0.9 | 50.6 |
| CDANN (2018)[60] | 56.3±2.0 | 47.1±0.9 | 57.2±1.1 | 42.4±0.8 | 50.8 |
| MLDG (2018)[61] | 59.2±0.1 | 49.0±0.9 | 58.4±0.9 | 41.4±1.0 | 52.0 |
| MMD (2018)[62] | 60.6±1.1 | 45.9±0.3 | 57.8±0.5 | 43.8±1.2 | 52.0 |
| IRM (2019)[63] | 56.5±2.5 | 49.8±1.5 | 57.1±2.2 | 38.6±1.0 | 50.5 |
| GroupDRO (2019)[64] | 60.4±1.5 | 48.3±0.4 | 58.6±0.8 | 42.2±0.8 | 52.4 |
| Mixup (2020)[65] | 67.6±1.8 | 51.0±1.3 | 59.0±N/A | 40.0±1.1 | 54.4 |
| ARM (2020)[66] | 60.1±1.5 | 48.3±1.6 | 55.3±0.6 | 40.9±1.1 | 51.2 |
| RSC (2020)[67] | 59.9±1.4 | 46.7±0.4 | 57.8±0.5 | 44.3±0.6 | 52.1 |
| SagNet (2021)[68] | 56.4±1.9 | 50.9±2.3 | 59.1±0.5 | 44.1±0.6 | 52.5 |
| V-REx (2021)[69] | 56.8±1.7 | 46.5±0.5 | 58.4±0.3 | 43.8±0.3 | 51.4 |
| AND-mask (2021)[70] | 54.7±1.8 | 48.4±0.5 | 55.1±0.5 | 41.3±0.6 | 49.8 |
| Fish (2021) [71] | - | - | - | - | 50.8 |
| SAND-mask (2021)[72] | 56.2±1.8 | 46.3±0.3 | 55.8±0.4 | 42.6±1.2 | 50.2 |
| Fishr (2022)[53] | 60.4±0.9 | 50.3±0.3 | 58.8±0.5 | 44.9±0.5 | 53.6 |
| GDM(2019)[48] | - | - | - | - | - |
| MTL(2021)[47] | 58.4±2.1 | 48.4±0.8 | 58.9±0.6 | 43.0±1.3 | 52.2 |
| Transfer(2021)[73] | - | - | - | - | - |
| Ood(2021)[49] | - | - | - | - | - |
| **MADG (ours)** | 60.0±1.2 | 51.8±0.2 | 57.4±0.3 | 45.6±0.5 | 53.7 |

(Theory-Based Methods label spans the GDM, MTL, Transfer, Ood rows)

Table A5: Accuracy (%) on each domain of the DomainNet dataset. We represent '-' for models which did not report accuracy for that domain. We also report deviation as ± N/A for models that did not report them.

| Algorithm | clip | info | paint | quick | real | sketch | Avg.(↑) |
|---|---|---|---|---|---|---|---|
| ERM (1998)[57] | 58.6±0.3 | 19.2±0.2 | 47.0±0.3 | 13.2±0.2 | 59.9±0.3 | 49.8±0.4 | 41.3 |
| CORAL (2016)[58] | 59.2±0.1 | 19.9±0.2 | 47.4±0.2 | 14.0±0.4 | 59.8±0.2 | 50.4±0.4 | 41.8 |
| DANN (2016)[59] | 53.1±0.2 | 18.3±0.1 | 44.2±0.7 | 11.9±0.1 | 55.5±0.4 | 46.8±0.6 | 38.3 |
| CDANN (2018)[60] | 54.6±0.4 | 17.3±0.1 | 44.2±0.7 | 12.8±0.2 | 56.2±0.4 | 45.9±0.5 | 38.5 |
| MLDG (2018)[61] | 59.3±0.1 | 19.6±0.2 | 46.8±0.2 | 13.4±0.2 | 60.1±0.4 | 50.4±0.3 | 41.6 |
| MMD (2018)[62] | 32.2±13.3 | 11.2±4.5 | 26.8±11.3 | 8.8±2.2 | 32.7±13.8 | 29.0±11.8 | 23.5 |
| IRM (2019)[63] | 40.4±6.6 | 12.1±2.7 | 31.4±5.7 | 9.8±1.2 | 37.7±9.0 | 36.7±5.3 | 28.0 |
| GroupDRO (2019)[64] | 47.2±0.5 | 17.5±0.4 | 34.2±0.3 | 9.2±0.4 | 51.9±0.5 | 40.1±0.6 | 33.4 |
| Mixup (2020)[65] | 55.6±0.1 | 18.7±0.4 | 45.1±0.5 | 12.8±0.3 | 57.6±0.5 | 48.2±0.4 | 39.6 |
| ARM (2020)[66] | 49.6±0.4 | 16.5±0.3 | 41.5±0.8 | 10.8±0.1 | 53.5±0.3 | 43.9±0.4 | 36.0 |
| RSC (2020)[67] | 55.0±1.2 | 18.3±0.5 | 44.4±0.6 | 12.5±0.1 | 55.7±0.7 | 47.8±0.9 | 38.9 |
| SagNet (2021)[68] | 57.7±0.3 | 19.1±0.1 | 46.3±0.5 | 13.5±0.4 | 58.9±0.4 | 49.5±0.2 | 40.8 |
| V-REx (2021)[69] | 43.3±4.5 | 14.1±1.8 | 32.5±5.0 | 9.8±1.1 | 43.5±5.6 | 37.7±4.5 | 30.1 |
| AND-mask (2021)[70] | 52.3±0.8 | 17.3±0.5 | 43.7±1.1 | 12.3±0.4 | 55.8±0.4 | 46.1±0.8 | 37.9 |
| Fish (2021)[71] | - | - | - | - | - | - | 43.4 |
| SAND-mask (2021)[72] | 43.8±1.3 | 15.2±0.2 | 38.2±0.6 | 9.0±0.2 | 47.1±1.1 | 39.9±0.6 | 32.2 |
| Fishr (2022)[53] | 58.3±0.5 | 20.2±0.2 | 47.9±0.2 | 13.6±0.3 | 60.5±0.3 | 50.5±0.3 | 41.8 |
| GDM(2019)[48] | - | - | - | - | - | | |
| MTL(2021)[47] | 58.0±0.4 | 19.2±0.2 | 46.2±0.1 | 12.7±0.2 | 59.9±0.1 | 49.0±0.0 | 40.8 |
| Transfer(2021)[73] | - | - | - | - | - | | |
| Ood(2021)[49] | - | - | - | - | - | | |
| **MADG (ours)** | 62.5±0.4 | 22.0±0.3 | 34.1±0.3 | 15.1±0.2 | 57.4±1.1 | 48.0±0.3 | 39.9 |

(Theory-Based Methods)

Table A6: Accuracy (%) on each domain of the VLCS dataset. We represent '-' for models which did not report accuracy for that domain. We also report deviation as ± N/A for models that did not report them.

| Algorithm | C | L | S | V | Avg.(↑) |
|---|---|---|---|---|---|
| ERM (1998) [57] | 97.6±0.3 | 67.9±0.7 | 70.9±0.2 | 74.0±0.6 | 77.6 |
| CORAL (2016) [58] | 97.3±0.2 | 67.5±0.6 | 71.6±0.6 | 74.5±N/A | 77.7 |
| DANN (2016) [59] | 99.0±0.2 | 66.3±1.2 | 73.4±1.4 | 80.1±0.5 | 79.7 |
| CDANN (2018) [60] | 98.2±0.1 | 68.8±0.5 | 74.3±0.6 | 78.1±0.5 | 79.9 |
| MLDG (2018) [61] | 97.1±0.5 | 66.6±0.5 | 71.5±0.1 | 75.0±0.9 | 77.5 |
| MMD (2018) [62] | 98.8±0.0 | 66.4±0.4 | 70.8±0.5 | 75.6±0.4 | 77.9 |
| IRM (2019) [63] | 97.3±0.2 | 66.7±0.1 | 71.0±2.3 | 72.8±0.4 | 76.9 |
| GroupDRO (2019) [64] | 96.7±0.2 | 65.9±0.2 | 72.8±0.8 | 73.4±1.3 | 77.4 |
| Mixup (2020) [65] | 97.8±0.4 | 67.2±0.4 | 71.5±0.2 | 75.7±0.6 | 78.1 |
| ARM (2020) [66] | 97.6±0.6 | 66.5±0.3 | 72.7±0.6 | 74.4±0.7 | 77.8 |
| RSC (2020) [67] | 98.0±0.4 | 67.2±0.3 | 70.3±1.3 | 75.6±0.4 | 77.8 |
| SagNet (2021) [68] | 97.4±0.3 | 66.4±0.4 | 71.6±0.1 | 75.0±0.8 | 77.6 |
| V-REx (2021) [69] | 98.4±0.2 | 66.4±0.7 | 72.8±0.1 | 75.0±1.4 | 78.1 |
| AND-mask (2021) [70] | 98.3±0.3 | 64.5±0.2 | 69.3±1.3 | 73.4±1.3 | 76.4 |
| Fish (2021) [71] | - | - | - | - | 77.8 |
| SAND-mask (2021) [72] | 97.6±0.3 | 64.5±0.6 | 69.7±0.6 | 73.0±1.2 | 76.2 |
| Fishr (2022) [53] | 97.6±0.7 | 67.3±0.5 | 72.2±0.9 | 75.7±0.3 | 78.2 |
| G2DM (2019)[48] | 95.5±N/A | 67.6 ±N/A | 69.4±N/A | 71.1±N/A | 75.9 |
| MTL(2021)[47] | 97.9±0.7 | 66.1±0.7 | 72.0±0.4 | 74.9±1.1 | 77.7 |
| Transfer(2021)[73] | - | - | - | - | - |
| Ood(2021)[49] | 97.8 ±N/A | 67.0±N/A | 69.5 ±N/A | 71±N/A | 76.3 |
| **MADG (ours)** | 98.5±0.2 | 65.8±0.3 | 73.1±0.3 | 77.3±0.1 | 78.7 |

(Theory-Based Methods)

**Results with Other Model Selection Strategies.** For completeness of analysis, we also study our method under the 'Train-domain' validation setting proposed in [35]. Although not our objective in this work, our method obtains competitive results with minimal hyperparameter search efforts, as shown in Tables A8 and A9.

Table A7: Accuracy (%) on each domain of the PACS dataset. We represent '-' for models which did not report accuracy for that domain. We also report deviation as ± N/A for models that did not report them.

| Algorithm | | P | A | C | S | Avg.(↑) |
|---|---|---|---|---|---|---|
| ERM (1998) [57] | | 96.2±0.3 | 86.5±1.0 | 81.3±0.6 | 82.7±1.1 | 86.7 |
| CORAL (2016) [58] | | 97.1±0.5 | 86.6±0.8 | 81.8±0.9 | 82.7±0.6 | 87.1 |
| DANN (2016) [59] | | 96.8±0.3 | 87.0±0.4 | 80.3±0.6 | 76.9±1.1 | 85.2 |
| CDANN (2018) [60] | | 97.3±0.4 | 87.7±0.6 | 80.7±1.2 | 77.6±1.5 | 85.8 |
| MLDG (2018) [61] | | 96.7±0.3 | 87.0±1.2 | 82.5±0.9 | 81.2±0.6 | 86.8 |
| MMD (2018) [62] | | 97.1±0.5 | 88.1±0.8 | 82.6±0.7 | 81.2±1.2 | 87.2 |
| IRM (2019) [63] | | 95.9±0.4 | 84.2±0.9 | 79.7±1.5 | 78.3±2.1 | 84.5 |
| GroupDRO (2019) [64] | | 97.1±0.3 | 87.5±0.5 | 82.9±0.6 | 81.1±1.2 | 87.1 |
| Mixup (2020)[65] | | 97.4±0.2 | 87.5 ±0.4 | 81.6±0.7 | 80.8±0.9 | 86.8 |
| ARM (2020)[66] | | 95.9±0.3 | 85.0±1.2 | 81.4±0.2 | 80.9±0.5 | 85.8 |
| RSC (2020)[67] | | 96.8±0.7 | 86.0±0.7 | 81.8±0.9 | 80.4±0.5 | 86.2 |
| SagNet (2021)[68] | | 96.3±0.8 | 87.4±0.5 | 81.2±1.2 | 80.7±1.1 | 86.4 |
| V-REx (2021)[69] | | 97.4±0.2 | 87.8±1.2 | 81.8±0.7 | 82.1±0.7 | 87.2 |
| AND-mask (2021)[70] | | 97.1±0.2 | 86.4±1.1 | 80.8±0.9 | 81.3±1.1 | 86.4 |
| Fish (2021) [71] | | - | - | - | - | 85.8 |
| SAND-mask (2021) [72] | | 97.1±0.3 | 86.1±0.6 | 80.3±1.0 | 80.0±1.3 | 85.9 |
| Fishr (2022) [53] | | 97.9±0.4 | 87.9±0.6 | 80.8±0.5 | 81.1±0.8 | 86.9 |
| G2DM (2019)[48] | Theory-Based Methods | 88.1 ±N/A | 66.6 ±N/A | 73.4 ±N/A | 66.19±N/A | 86.7 |
| MTL(2021) [47] | | 96.5±0.7 | 87.0±0.2 | 82.7±0.8 | 80.5±0.8 | 86.7 |
| Transfer(2021)[73] | | - | - | - | - | 85.3 |
| Ood(2021)[49] | | 96.8±N/A | 88.7±N/A | 81.7±N/A | 79.0±N/A | 86.6 |
| **MADG (ours)** | | 97.7±0.3 | 87.8±0.5 | 82.2±0.6 | 78.3±0.4 | 86.5 |

Table A8: Accuracy (%) on each domain of the OfficeHome dataset using alternate hyperparameter selection method. We represent '-' for models which did not report accuracy for that domain. We also report deviation as ± N/A for models that did not report them.

| Algorithm | | A | C | P | R | Avg.(↑) |
|---|---|---|---|---|---|---|
| ERM (1998)[57] | | 61.3±0.7 | 52.4±0.3 | 75.8±0.1 | 76.6±0.3 | 66.5 |
| CORAL (2016)[58] | | 65.3±0.4 | 54.4±0.5 | 76.5±0.1 | 78.4±0.5 | 68.7 |
| DANN (2016)[59] | | 59.9±1.3 | 53.0±0.3 | 73.6±0.7 | 76.9±0.5 | 65.9 |
| CDANN (2018)[60] | | 61.5±1.4 | 50.4±2.4 | 74.4±0.9 | 76.6±0.8 | 65.8 |
| MLDG (2018)[61] | | 61.5±0.9 | 53.2±0.6 | 75.0±1.2 | 77.5±0.4 | 66.8 |
| MMD (2018)[62] | | 60.4±0.2 | 53.3±0.3 | 74.3±0.1 | 77.4±0.6 | 66.3 |
| IRM (2019)[63] | | 58.9±2.3 | 52.2±1.6 | 72.1±2.9 | 74.0±2.5 | 64.3 |
| GroupDRO (2019)[64] | | 60.4±0.7 | 52.7±1.0 | 75.0±0.7 | 76.0±0.7 | 66.0 |
| Mixup (2020)[65] | | 62.4±0.8 | 54.8±0.6 | 76.9±0.3 | 78.3±0.2 | 68.1 |
| ARM (2020)[66] | | 58.9±0.8 | 51.0±0.5 | 74.1±0.1 | 75.2±0.3 | 64.8 |
| RSC (2020)[67] | | 60.7±1.4 | 51.4±0.3 | 74.8±1.1 | 75.1±1.3 | 65.5 |
| SagNet (2021)[68] | | 63.4±0.2 | 54.8±0.4 | 75.8±0.4 | 78.3±0.3 | 68.1 |
| V-REx (2021)[69] | | 60.7±0.9 | 53.0±0.9 | 75.3±0.1 | 76.6±0.5 | 66.4 |
| AND-mask (2021)[70] | | 59.5±1.2 | 51.7±0.2 | 73.9±0.4 | 77.1±0.2 | 65.6 |
| Fish (2021) [71] | | - | - | - | - | 68.6 |
| SAND-mask (2021)[72] | | 60.3±0.5 | 53.3±0.7 | 73.5±0.7 | 76.2±0.3 | 65.8 |
| Fishr (2022)[53] | | 62.4±0.5 | 54.4±0.4 | 76.2±0.5 | 78.3±0.1 | 67.8 |
| G2DM (2019)[48] | Theory-Based Methods | - | - | - | - | - |
| MTL(2021) [47] | | 61.5±0.7 | 52.4±0.6 | 74.9±0.4 | 76.8±0.4 | 66.4 |
| Transfer(2021)[73] | | - | - | - | - | - |
| Ood(2021)[49] | | 61.9 ±N/A | 55.6 ±N/A | 74.7 ±N/A | 76.3 ±N/A | 67.1 |
| **MADG (ours)** | | 67.6±0.2 | 54.1±0.3 | 78.4±0.3 | 80.3±0.5 | 70.1 |

Table A9: Accuracy (%) on each domain of the PACS dataset using alternate hyperparameter selection method. We represent '-' for models which did not report accuracy for that domain. We also report deviation as ± N/A for models that did not report them.

| Algorithm | | P | A | C | S | Avg.(↑) |
|---|---|---|---|---|---|---|
| ERM (1998)[57] | | 97.2±0.3 | 84.7±0.4 | 80.8±0.6 | 79.3±1.0 | 85.5 |
| CORAL (2016)[58] | | 97.5±0.3 | 88.3±0.2 | 80.0±0.5 | 78.8±1.3 | 86.2 |
| DANN (2016)[59] | | 97.3±0.4 | 86.4±0.8 | 77.4±0.8 | 73.5±2.3 | 83.6 |
| CDANN (2018)[60] | | 96.8±0.3 | 84.6±1.8 | 75.5±0.9 | 73.5±0.6 | 82.6 |
| MLDG (2018)[61] | | 97.4±0.3 | 85.5±1.4 | 80.1±1.7 | 76.6±1.1 | 84.9 |
| MMD (2018)[62] | | 96.6±0.2 | 86.1±1.4 | 79.4±0.9 | 76.5±1.5 | 84.6 |
| IRM (2019)[63] | | 96.7±0.6 | 84.8±1.3 | 76.4±1.1 | 76.1±1.0 | 83.5 |
| GroupDRO (2019)[64] | | 96.7±0.3 | 83.5±0.9 | 79.1±0.6 | 78.3±2.0 | 84.4 |
| Mixup (2020)[65] | | 97.6±0.1 | 86.1 ±0.5 | 78.9±0.8 | 75.8±1.8 | 84.6 |
| ARM (2020)[66] | | 97.4±0.3 | 86.8±0.6 | 76.8±0.5 | 79.3±1.2 | 85.1 |
| RSC (2020)[67] | | 97.6±0.3 | 85.4±0.8 | 79.7±1.8 | 78.2±1.2 | 85.2 |
| SagNet (2021)[68] | | 97.1±0.1 | 87.4±1.0 | 80.7±0.6 | 80.0±0.4 | 86.3 |
| V-REx (2021)[69] | | 96.9±0.5 | 86.0±1.6 | 79.1±0.6 | 77.7±1.7 | 84.9 |
| AND-mask (2021)[70] | | 96.9±0.4 | 85.3±1.4 | 79.2±2.0 | 76.2±1.4 | 84.4 |
| Fish (2021)[71] | | - | - | - | - | 85.5 |
| SAND-mask (2021)[72] | | 96.3±0.2 | 85.8±1.7 | 79.2±0.8 | 76.9±2.0 | 84.6 |
| Fishr (2022)[53] | | 97.0±0.1 | 88.4±0.2 | 78.7±0.7 | 77.8±2.0 | 85.5 |
| G2DM (2019)[48] | Theory-Based Methods | - | - | - | - | - |
| MTL(2021)[47] | | 96.4±0.8 | 87.5±0.8 | 77.1±0.5 | 77.3±1.8 | 84.6 |
| Transfer(2021)[73] | | - | - | - | - | - |
| Ood(2021)[49] | | 96.2 ±N/A | 85.2 ±N/A | 80.4 ±N/A | 77.7±N/A | 84.9 |
| **MADG (ours)** | | 97.6±0.7 | 84.9±0.5 | 80.6±0.6 | 76.7±0.6 | 85.0 |

**Results with additional baselines.** In this work, we also compare the proposed **MADG** with a recent algorithm SD [74], which reported results on datasets different from ours. For a fair comparison, we run the SD model on the VLCS, PACS, and TerraIncognita datasets. As seen from Tables A10, A11 and A12, the proposed **MADG** model achieves better performance on all the domains across all three datasets.

Table A10: Accuracy(%) on VLCS dataset. Avg. = Average accuracy

| Algorithm | V | C | L | S | Avg.(↑) |
|---|---|---|---|---|---|
| SD(2021)[74] | 72.6 ±1.7 | 91.4±1.7 | 64.1±1.6 | 63.3±1.2 | 72.3 |
| **MADG (ours)** | **77.3 ±0.1** | **98.5±0.2** | **65.8±0.3** | **73.1±0.3** | **78.7** |

Table A11: Accuracy(%) on PACS dataset. Avg. = Average accuracy

| Algorithm | A | C | P | S | Avg.(↑) |
|---|---|---|---|---|---|
| SD (2021)[74] | 82.7 ±0.9 | 77.4±3 | 97.1±0.7 | 68.5±2.4 | 81.4 |
| **MADG (ours)** | **87.8±0.5** | **82.2±0.6** | **97.7±0.3** | **78.3±0.4** | **86.5** |

## A5  Adapting MADG to Multi-source DA

As stated earlier, our Theorem 1 can be viewed as an upper bound for error in the multi-source domain adaptation (DA) setting (which is new by itself too). While this is not our primary focus in this work, we report preliminary empirical results for a multi-source DA algorithm that reduces the first two terms in Theorem 1. We approximate the source errors and the MDD loss between labeled

Table A12: Accuracy(%) on TerraIncognita dataset. Avg. = Average accuracy

| Algorithm | L100 | L46 | L43 | L38 | Avg.($\uparrow$) |
|---|---|---|---|---|---|
| SD (2021)[74] | 39.5 $\pm$8.7 | 41.4 $\pm$1.2 | 49.2 $\pm$1 | 34.1$\pm$6.2 | 41.1 |
| **MADG (ours)** | **60.0 $\pm$1.2** | **45.6$\pm$0.5** | **57.4$\pm$0.3** | **51.8$\pm$0.2** | **53.7** |

and unlabeled source domains as shown in Eqn (A5.1) below.

$$\mathcal{E}(\hat{D}_{\mathcal{S}_i}) = \mathbb{E}_{(x,y)\sim\hat{D}_{\mathcal{S}_i}}\Big[L\Big(f\big(\mathcal{G}(x)\big), y\Big)\Big]$$

$$\mathcal{D}_{(\hat{\rho},l)}\big(\hat{D}_{\mathcal{S}_i}, \hat{D}_{\mathcal{S}_T}\big) = \mathbb{E}_{(x,y)\sim\hat{D}_{\mathcal{S}_T}}\Big[L'\Big(f'_l\big(\mathcal{G}(x)\big), f\big(\mathcal{G}(x)\big)\Big)\Big] - \hat{\rho}\mathbb{E}_{(x,y)\sim\hat{D}_{\mathcal{S}_i}}\Big[L\Big(f'_l\big(\mathcal{G}(x)\big), f\big(\mathcal{G}(x)\big)\Big)\Big]$$
(A5.1)

where $l = \{1, \ldots, N_s - 1\}$. The final optimization problem can then be given as below in Eqn A5.2.

$$\min_{f,\mathcal{G}} \sum_{i=1}^{N_s-1} \alpha_i\Big(\mathcal{E}(\hat{D}_{\mathcal{S}_i})\Big) + \sum_{l=1}^{N_s-1}\Big(\mathcal{D}_{(\hat{\rho},l)}\big(\hat{D}_{\mathcal{S}_i}, \hat{D}_{\mathcal{S}_T}\big)\Big), \quad \max_{f'_1,\ldots,f'_j} \sum_{l=1}^{Ns-1}\Big(\mathcal{D}_{(\hat{\rho},l)}\big(\hat{D}_{\mathcal{S}_i}, \hat{D}_{\mathcal{S}_T}\big)\Big)$$
(A5.2)

Algorithm A1 summarizes the methodology to adapt MADG to the multi-source DA setting. The proposed multi-source DA (MADA) algorithm follows a similar structure to MADG, where we train the adversarial model in two steps. In the first step, we update the parameters $f$ and $\mathcal{G}$. In the next step, we do a forward pass to compute the outputs from $f$ and then update the parameters $f'$ and $\mathcal{G}$. We evaluate the MADA algorithm's performance on the Office Caltech dataset. We use ResNet101 [54] architecture pre-trained on the ImageNet dataset as the feature extractor and train the model using stochastic gradient descent with a momentum optimizer. The results on Office Caltech are reported in Table A13. As seen from Table A13, MADA outperforms previous models across most domains and achieves the best average accuracy. It improves significantly, 1.2% average accuracy, compared to MDAN, a theory-inspired model for the multi-source DA problem.

---

**Algorithm A1** Margin-based adversarial learning for Multi-source Domain Adaptation (MADA)

---

**Require:** $N_s$ labeled source domains, $(x, y) \sim \mathcal{D}_{\mathcal{S}_i}$

  **for** $a \leftarrow 1$ to total_epochs **do**

    **for** $b \leftarrow 1$ to total_batches **do**

      Update parameters according to $\min_{f,\mathcal{G}} \sum_{i=1}^{N_s-1} \alpha_i\mathbb{E}_{(x,y)\sim\hat{\mathcal{D}}_{\mathcal{S}_i}} L(f(\mathcal{G}(x)), y)$

      $\hat{y}_{\mathcal{D}_{\mathcal{S}_i}} \leftarrow f(\mathcal{G}(x)$ for $i = \{1, \ldots, N_s - 1\}$

      Calculate $MDD_l$ according to equation (A5.1) and using $\hat{y}_{\mathcal{D}_{\mathcal{S}_i}}$ for $l = 1, \ldots, N_s - 1$

      Calculate Transfer loss $= \sum_{l=1}^{N_s-1}(\mathcal{D}_{(\hat{\rho},l)}(\hat{\mathcal{D}}_{\mathcal{S}_i}, \hat{\mathcal{D}}_{\mathcal{S}_T}))$

      Update parameters, $f'$ and $\mathcal{G}$ according to equation (A5.2)

    **end for**

  **end for**

---

## A6  More Analysis

This section shows further empirical analysis conducted on the proposed MADG model.

**MDD formulation** The proposed MADG algorithm uses margin-based discrepancy, MDD, to compute disagreements between classifiers for a pair of domains, as restated below.

$$\mathrm{d}_{f,\mathcal{F}}^{(\rho)}\big(\mathcal{D}_{S_i}, \mathcal{D}_{S_k}\big) \triangleq \sup_{f'\in\mathcal{F}}\Big(\mathrm{disp}_{\mathcal{D}_{S_k}}^{(\rho)}\big(f', f\big) - \mathrm{disp}_{\mathcal{D}_{S_i}}^{(\rho)}\big(f', f\big)\Big)$$

We reformulate the MDD (called MDD-new) to compute disagreements for multiple domains simultaneously, as shown below.

$$\mathrm{d}_{f,\mathcal{F}}^{(\rho)}\Big(\mathcal{D}_{S_i}, \mathcal{D}_{S_k}, \mathcal{D}_{S_p}\Big) \triangleq \sup_{f'\in\mathcal{F}}\Big(\mathrm{disp}_{\mathcal{D}_{S_p}}^{(\rho)}\big(f', f\big) - \mathrm{disp}_{\mathcal{D}_{S_k}}^{(\rho)}\big(f', f\big) - \mathrm{disp}_{\mathcal{D}_{S_i}}^{(\rho)}\big(f', f\big)\Big)$$

Table A13: Accuracy(%) of the proposed MADA model on Office Caltech dataset. We report deviation as ±N/A for models that did not report them. A = Amazon domain, C = Caltech domain, D = DSLR domain, W = Webcam domain, Avg. = Average accuracy

| Algorithm | A | C | D | W | Avg.($\uparrow$) |
|---|---|---|---|---|---|
| DANN (2016)[A29] | 94.8±N/A | 89.7±N/A | 98.2±N/A | 99.3±N/A | 95.5 |
| MDAN (2018)[A30] | 95.4±N/A | 91.8±N/A | 98.6 ±N/A | 98.9±N/A | 96.1 |
| $M^3$SDA (2019)[A31] | 94.5±N/A | 92.2±N/A | 99.2±N/A | 99.5±N/A | 96.4 |
| MDDA (2020)[A32] | 95.5 ±0.2 | 91.6±0.3 | 99.0±0.0 | 99.42±0.2 | 96.4 |
| DARN (2020)[A33] | 95.7±0.4 | 91.8 ±0.3 | 99.4 .1 | 99.3 ±0.2 | 96.5 |
| CMSS (2020)[A34] | 96.0 ±N/A | 93.7±N/A | 99.3±N/A | 99.6±N/A | 97.2 |
| MIAN (2021)[A35] | 96.1 ±0.1 | **94.6** ±0.1 | 99.0 ±0.1 | 99.32 ±0.2 | 97.24 |
| **MADA (ours)** | **96.3**±0.1 | 93.4±0.2 | **100.0**±0.1 | **99.7**±0.1 | **97.3** |

We compare the performance between our original MDD and new MDD formulation on the PACS dataset and report it in Table A14. The model trained with the original MDD formulation achieves better accuracy across all domains as the pairwise disparity is able to capture better domain discrepancy and hence learn better domain-invariant features. More effective implementations of the multi-domain MDD strategy can be an interesting direction of future work.

Table A14: Accuracy(%) of the MADG algorithm with different MDD formulations. Org.=Original MDD, New=MDD-new

| | Org. | New |
|---|---|---|
| P | **97.7** | 97.5 |
| A | **87.8** | 87.4 |
| C | **82.2** | 81.2 |
| S | **78.3** | 76.6 |
| Avg | **86.5** | 85.6 |

**Multistep training.** As mentioned in Algorithm 1 and Section 5.2 in the main paper, we solve the minimax problem by training the MADG model in two crucial steps instead of a joint update in a single step. We report in Table A15 that such two-step training improves the performance of the model across all the datasets. As seen in Table A15, we also compare with the 'j' step training where each $f'_l$ classifier along with the feature extractor ($\mathcal{G}$) is updated sequentially. The two-step training methodology outperforms the 'j' step methodology across all datasets. This is because in the two-step methodology, the parameters $\mathcal{G}$ and $f$ are first updated to learn features pertaining to the classification task. These updated parameters generate better pseudo labels to train the $f'$ parameters in the next step. This decoupled updation of parameters is not possible in joint-update methodology. Similarly, for the j-step methodology, after the first two updates of $\mathcal{G}$, the feature extractor's efficiency in extracting features relevant to the classification task decrease as it receives only gradients from the MDD loss for the next '$j - 2$' steps.

Table A15: Accuracy(%) of the MADG model for different multi-step training.

| Dataset | Two | Single | jstep |
|---|---|---|---|
| VLCS | **78.7** | 78.4 | 77.9 |
| PACS | **86.5** | 83.0 | 81.9 |
| OH | **71.3** | 70.9 | 69.7 |
| TI | **53.7** | 52.4 | 52.7 |

## A7 Limitations and Future Directions

The proposed model learns invariant features via adversarial learning. Although the minimax method generates good invariant features, it can be further enhanced using explainable models such as GradCAM [A36]. These explainable models help identify features that contribute to the end task and are domain-invariant, thus being capable of boosting the MADG model's performance. In this work, we also consider a convex combination of multiple sources and their respective mixture weights ($\pi_i, \forall i = \{1, \ldots, N_s\}$). These mixture weights can be further improved by identifying source domains that are harder to align and assigning more weight to such domains. The proposed model also requires domain labels during training, similar to recent work such as [53][71]

## Additional References

[A1] D. Li, Y. Yang, Y.-Z. Song, and T. Hospedales, "Learning to generalize: Meta-learning for domain generalization," in Proceedings of the AAAI conference on artificial intelligence, vol. 32, no. 1, 2018.

[A2] Y. Balaji, S. Sankaranarayanan, and R. Chellappa, "Metareg: Towards domain generalization using meta-regularization," Advances in neural information processing systems, vol. 31, 2018.

[A3] D. Li, J. Zhang, Y. Yang, C. Liu, Y.-Z. Song, and T. M. Hospedales, "Episodic training for domain generalization," in Proceedings of the IEEE/CVF International Conference on Computer Vision, 2019, pp. 1446–1455.

[A4] Y. Zhao, Z. Zhong, F. Yang, Z. Luo, Y. Lin, S. Li, and N. Sebe, "Learning to generalize unseen domains via memory-based multi-source meta-learning for person re-identification," in Proceedings of the IEEE/CVF Conference on Computer Vision and Pattern Recognition, 2021, pp. 6277–6286.

[A5] S. Choi, T. Kim, M. Jeong, H. Park, and C. Kim, "Meta batch-instance normalization for generalizable person re-identification," in Proceedings of the IEEE/CVF conference on Computer Vision and Pattern Recognition, 2021, pp. 3425–3435.

[A6] Q. Dou, D. Coelho de Castro, K. Kamnitsas, and B. Glocker, "Domain generalization via model-agnostic learning of semantic features," Advances in Neural Information Processing Systems, vol. 32, 2019.

[A7] K. Zhou, Y. Yang, Y. Qiao, and T. Xiang, "Mixstyle neural networks for domain generalization and adaptation," arXiv preprint arXiv:2107.02053, 2021.

[A8] R. Volpi and V. Murino, "Addressing model vulnerability to distributional shifts over image transformation sets," in Proceedings of the IEEE/CVF International Conference on Computer Vision, 2019, pp. 7980–7989.

[A9] G. Blanchard, A. A. Deshmukh, Ü. Dogan, G. Lee, and C. Scott, "Domain generalization by marginal transfer learning," The Journal of Machine Learning Research, vol. 22, no. 1, pp. 46–100, 2021.

[A10] F. Qiao, L. Zhao, and X. Peng, "Learning to learn single domain generalization," in Proceedings of the IEEE/CVF Conference on Computer Vision and Pattern Recognition, 2020, pp. 12 556–12 565.

[A11] N. Somavarapu, C.-Y. Ma, and Z. Kira, "Frustratingly simple domain generalization via image stylization," arXiv preprint arXiv:2006.11207, 2020.

[A12] Y. Ruan, Y. Dubois, and C. J. Maddison, "Optimal representations for covariate shift," in The Tenth International Conference on Learning Representations, ICLR, 2022.

[A13] I. Albuquerque, N. Naik, J. Li, N. Keskar, and R. Socher, "Improving out-of-distribution generalization via multi-task self-supervised pretraining," arXiv preprint arXiv:2003.13525, 2020.

[A14] S. Harary, E. Schwartz, A. Arbelle, P. Staar, S. Abu-Hussein, E. Amrani, R. Herzig, A. Alfassy, R. Giryes, H. Kuehne et al., "Unsupervised domain generalization by learning a bridge across domains," in Proceedings of the IEEE/CVF Conference on Computer Vision and Pattern Recognition, 2022, pp. 5280–5290.

[A15] X. Zhang, L. Zhou, R. Xu, P. Cui, Z. Shen, and H. Liu, "Towards unsupervised domain generalization," in Proceedings of the IEEE/CVF Conference on Computer Vision and Pattern Recognition, 2022, pp. 4910–4920.

[A16] F.-E. Yang, Y.-C. Cheng, Z.-Y. Shiau, and Y.-C. F. Wang, "Adversarial teacher-student representation learning for domain generalization," Advances in Neural Information Processing Systems, vol. 34, pp. 19 448–19 460, 2021.

[A17] S. Jeon, K. Hong, P. Lee, J. Lee, and H. Byun, "Feature stylization and domain-aware contrastive learning for domain generalization," in Proceedings of the 29th ACM International Conference on Multimedia, 2021, pp. 22–31.

[A18] S. Bucci, A. D'Innocente, Y. Liao, F. M. Carlucci, B. Caputo, and T. Tommasi, "Self-supervised learning across domains," IEEE Transactions on Pattern Analysis and Machine Intelligence, vol. 44, no. 9, pp. 5516–5528, 2021.

[A19] G. Blanchard, G. Lee, and C. Scott, "Generalizing from several related classification tasks to a new unlabeled sample," Advances in neural information processing systems, vol. 24, 2011.

[A20] D. Li, H. Gouk, and T. Hospedales, "Finding lost dg: Explaining domain generalization via model complexity," arXiv preprint arXiv:2202.00563, 2022.

[A21] M. Zhang, H. Marklund, N. Dhawan, A. Gupta, S. Levine, and C. Finn, "Adaptive risk minimization: Learning to adapt to domain shift," Advances in Neural Information Processing Systems, vol. 34, pp. 23 664–23 678, 2021.

[A22] A. Nasery, S. Thakur, V. Piratla, A. De, and S. Sarawagi, "Training for the future: A simple gradient interpolation loss to generalize along time," Advances in Neural Information Processing Systems, vol. 34, pp. 19 198–19 209, 2021.

[A23] J. Cha, S. Chun, K. Lee, H.-C. Cho, S. Park, Y. Lee, and S. Park, "Swad: Domain generalization by seeking flat minima," Advances in Neural Information Processing Systems, vol. 34, pp. 22 405–22 418, 2021.

[A24] A. Robey, G. J. Pappas, and H. Hassani, "Model-based domain generalization," Advances in Neural Information Processing Systems, vol. 34, pp. 20 210–20 229, 2021.

[A25] J. Cha, K. Lee, S. Park, and S. Chun, "Domain generalization by mutual-information regularization with pre-trained models," in Proceedings of the European Conference on Computer Vision (ECCV), 2022, pp. 440–457.

[A26] Y. Wald, A. Feder, D. Greenfeld, and U. Shalit, "On calibration and out-of-domain generalization," Advances in neural information processing systems, vol. 34, pp. 2215–2227, 2021.

[A27] D. Teney, E. Abbasnejad, S. Lucey, and A. Van den Hengel, "Evading the simplicity bias: Training a diverse set of models discovers solutions with superior ood generalization," in Proceedings of the IEEE/CVF Conference on Computer Vision and Pattern Recognition, 2022, pp. 16 761–16 772.

[A28] A. D'Amour, K. Heller, D. Moldovan, B. Adlam, B. Alipanahi, A. Beutel, C. Chen, J. Deaton, J. Eisenstein, M. D. Hoffman et al., "Underspecification presents challenges for credibility in modern machine learning," The Journal of Machine Learning Research, vol. 23, no. 1, pp. 10 237–10 297, 2022.

[A29] Y. Ganin, E. Ustinova, H. Ajakan, P. Germain, H. Larochelle, F. Laviolette, M. Marchand, and V. Lempitsky, "Domain-adversarial training of neural networks," The journal of machine learning research, vol. 17, no. 1, pp. 2096–2030, 2016.

[A30] H. Zhao, S. Zhang, G. Wu, J. M. Moura, J. P. Costeira, and G. J. Gordon, "Adversarial multiple source domain adaptation," Advances in neural information processing systems, vol. 31, 2018.

[A31] X. Peng, Q. Bai, X. Xia, Z. Huang, K. Saenko, and B. Wang, "Moment matching for multi-source domain adaptation," in Proceedings of the IEEE/CVF international conference on computer vision, 2019, pp. 1406–1415.

[A32] S. Zhao, G. Wang, S. Zhang, Y. Gu, Y. Li, Z. Song, P. Xu, R. Hu, H. Chai, and K. Keutzer, "Multi-source distilling domain adaptation," in Proceedings of the AAAI Conference on Artificial Intelligence, vol. 34, no. 07, 2020, pp. 12 975–12 983.

[A33] J. Wen, R. Greiner, and D. Schuurmans, "Domain aggregation networks for multi-source domain adaptation," in International Conference on Machine Learning.   PMLR, 2020, pp. 10 214–10 224.

[A34] L. Yang, Y. Balaji, S.-N. Lim, and A. Shrivastava, "Curriculum manager for source selection in multi-source domain adaptation," in Computer Vision–ECCV 2020: 16th European Conference, Glasgow, UK, August 23–28, 2020, Proceedings, Part XIV 16.   Springer, 2020, pp. 608–624.

[A35] G. Y. Park and S. W. Lee, "Information-theoretic regularization for multi-source domain adaptation," in Proceedings of the IEEE/CVF International Conference on Computer Vision, 2021, pp. 9214–9223.

[A36] R. R. Selvaraju, M. Cogswell, A. Das, R. Vedantam, D. Parikh, and D. Batra, "Grad-cam: Visual explanations from deep networks via gradient-based localization," in 2017 IEEE International Conference on Computer Vision (ICCV), 2017, pp. 618–626.