# OpenReview forum: "MADG: Margin-based Adversarial Learning for Domain Generalization"
_NeurIPS.cc/2023/Conference — NeurIPS 2023 poster_

### Official Review · Reviewer_zZgJ · 2023-06-29

**Soundness:** 2 fair
**Presentation:** 2 fair
**Contribution:** 2 fair
**Rating:** 5
**Confidence:** 3

**Summary:**

This paper aims to ease the distribution problem from a theoretical perspective, which uses margin loss and a scoring function to describe the relationship between domains, and the generalization bound in terms of functional class complexity is subsequently analyzed. Based on their theoretical analysis, a margin-based adversarial framework, which is developed upon the classical DANN method, is further proposed. Results conducted in the Domanbed benchmark show their results are competitive among existing arts.

**Strengths:**

1. It is always good to see some theoretical analysis for DG.

2. Experiments will the five datasets are appreciated.


**Weaknesses:**

1. The motivation in the introduction is problematic. The literature is happy to see inspiring theory, but the theory itself is not the purpose. Rather, the paper should emphasize what part of existing work requires a proper theoretical explanation. After all, how to solve domain generalization is the problem. The authors may consider revise the introduction part.

2. The idea of adversarial training for DG (either DANN [77] or MMD [79]), is shown to be less effective than ERM according to different benchmarks [32, a]. However, this work shows significantly better results than ERM, what is the advantage of the proposed MADG compared with them?

3. According to Line 282, and the implementation details in the supplementary material, I think the comparisons are unfair, as the Domainbed uses randomly selected hyperparameters (batch size, learning rate, etc.), which are fixed in their experiments.

4. More effective approaches should be compared, such as SD [b], and Miro [c], and it is suggested to reevaluate their code (at least some of them) on the same device.

[a] OoD-Bench: Quantifying and Understanding Two Dimensions of Out-of-Distribution Generalization, in CVPR'21.

[b] Gradient Starvation: A Learning Proclivity in Neural Networks, in NeurIPS'21.

[c] MIRO: Mutual Information Regularization with Oracle, in ECCV'22.

**Questions:**

See weakness.

**Limitations:**

It seems domain labels must be available during training, this should be listed as a limitation as domain labels are not always available.

---

> ### Author Rebuttal · Authors · 2023-08-09
>
> We thank the reviewer for the valuable comments and suggestions. We respond below to each of the concerns/suggestions.
>
> > The motivation in the introduction is problematic. The literature is happy to see inspiring theory, but the theory itself is not the purpose. Rather, the paper should emphasize what part of existing work requires a proper theoretical explanation. After all, how to solve domain generalization is the problem. The authors may consider revise the introduction part.
> >
> **Response:** We apologize for not being clearer on this. Our overall intention was to suggest that having strong theoretical underpinnings along with improved performance is an added advantage for a method. We will  revisit the phraseage and temper/substantiate our claim on motivating our method using theoretical framework as suggested. As discussed in Sec 1 (L39-42), our motivation in this work was to develop an adversarial DG algorithm which uses a disparaity discrepancy that is based on margin loss because of its advantages as discussed in Sec 4 (L129-144. We also develop a generalization bound for an unseen domain based on margin loss and scoring functions, which are more informative than 0-1 loss based bounds.
>
> >The idea of adversarial training for DG (either DANN [77] or MMD [79]), is shown to be less effective than ERM according to different benchmarks [32, a]. However, this work shows significantly better results than ERM, what is the advantage of the proposed MADG compared with them?
> >
> **Response:** As discussed in Sec 4 (L136-139), one of the important advantages of the margin-based discrepancy used in MADG (defined in Eq (5) and (6)) is that it considers the classifier f (that is used to perform the task) in its formulation. The adversarial methods in previous works have used discrepancy measures such as MMD [79] (moment matching metric) and DANN [77] (based on H$\Delta$H discrepancy) that do not consider the task (the classifier) in their formulations. Also as discused in Sec 4 (L132-144), margin-based discrepancy is tighter and can be efficiently optimized as compared to $H\Delta H$ discrepancy (ours computes the supremum over just one variable, whereas the latter computes over two variables). We hypothesize that these reasons make the margin-based disparity measure used in MADG perform better domain alignment and learn better domain-invariant features for the DG task.
>
> > According to Line 282, and the implementation details in the supplementary material, I think the comparisons are unfair, as Domainbed uses randomly selected hyperparameters (batch size, learning rate, etc.), which are fixed in their experiments.
> >
> **Response:** In DomainBed, hyperparameter tuning is done by selecting a hyperparameter value from a range of values randomly drawn from a distribution as shown in Table 6 in [32]. We follow the same approach. In Table A1 in our appendix, we report the hyperparameter values of the MADG model for each dataset. Simiarl to DomainBed, we employ hyperparameter tuning from among a range of values for each parameter, as detailed in the below Table 1. We will include these, and clarify this in the Appendix.
>
> Table 1: Ranges of values for hyperparameter tuning in our MADG model
> | Hyperparameter | Search Values |
> | -------- | -------- |
> | Margin     | [1, 1.5, 2, 3]     |
> | Learning rate | [0.01, 0.04, 0.001, 0.004, 0.0001] |
> | Momentum | [0.85, 0.9, 0.95] |
> | Weight Decay | [0.005, 0.0005] |
> | Dropout | [0.3, 0.4, 0.45, 0.5] |
> | Learning rate decay | [0.001, 0.0002] |
>
> >More effective approaches should be compared, such as SD [b], and Miro [c], and it is suggested to reevaluate their code (at least some of them) on the same device.
> >
> **Response:**  As discussed in Sec 6 (Lines 278-280) we evaluate our model on five different benchmark datasets (OfficeHome, PACS, VLCS, TerraIncognita and DomainNet) that is followed by recent papers [71,88,89] and DomainBed benchmark [32]. To the best of our knowledge, this is comprehensive for DG work. The paper [b] did not report results on these datasets, however, as suggested, we run the SD model [b] on OfficeHome dataset for three trials and report the average results below in Table 2. As seen from Table 2, MADG (our model) outperforms SD model on all domains.
>
> Also, as stated in Appendix A2 (L85-86), we follow recent state-of-the-art work [71] (also supported by recent papers [A1-A3]) in their test-domain methodology for hyperparameter selection. The paper [c] proposes the MIRO model and reports results only for training-domain validation-based model selection setting. For fair comparision, based on your suggestion, we run the MIRO model for three trials on our setting. The results are reported below in Table 2 where we can see that the average accuracy of our model outperforms the MIRO model. The MADG model also outperforms MIRO on two domains and achieves the same accuracy on the \`Real World\`domain.
>
> Table 2: Accuracy (%) on OfficeHome dataset
> |Model|Art|Clipart|Product|Real World| Average|
> |-------|--------|-------|-------|------|------|
> |Miro[c]|67($\pm0.7$)|56.5($\pm0.9$)|79.4($\pm1.4$)|81.5 ($\pm0.5$)|71.1|
> |SD [b]|64.8($\pm0.9$)|49.9($\pm1.2$)|75.6($\pm1.3$)|79.1($\pm0.2$)|67.4
> |MADG (ours)| 68.6($\pm0.5$)|55.5($\pm 0.2$)|79.6($\pm0.3$)|81.5($\pm0.4$)|71.3|
>
> > It seems domain labels must be available during training, this should be listed as a limitation as domain labels are not always available.
> >
> **Response:** As with recent state-of-the-art DG methods [71],[88], our method requires the knowledge of domain assignment of each data point. We will be happy to state this assumption/limitation explicitly.
>
> [a] OoD-Bench: Quantifying and Understanding Two Dimensions of Out-of-Distribution Generalization, in CVPR'21.
> [b] Gradient Starvation: A Learning Proclivity in Neural Networks, in NeurIPS'21.
> [c] MIRO: Mutual Information Regularization with Oracle, in ECCV'22.

---

> > ### Comment · Reviewer_zZgJ · 2023-08-14
> > **Thanks for the response**
> >
> > Thanks for the rebuttal, some of my concerns are addressed. But my main concern regarding the hyper-parameter setting, which I think leads to unfair comparisons, remains. In Table 8 (not Table 6) in [32], the parameter for each trial is randomly selected in a uniform distribution, and the performance is averaged from different trials. That being said, the parameters are different for different trials, and simply selecting the best-performing parameter for all trials violates the random selecting rule. Besides, different datasets use the same set of hyper-parameter in [32], which are also reported differently in Table A1 in the supp. material. For this reason, I cannot recommend acceptance for this work.

---

> > > ### Author Response · Authors · 2023-08-14
> > >
> > > We thank the reviewer for acknowledging our response, and are glad to know that it helped resolve concerns.
> > >
> > > On the pending issue, we'd like to clarify any misunderstanding. Firstly, on a minor note, we meant Table 6 in the ICLR version of [32] (the version available on OpenReview), which is Table 8 of its arxiv version [32].
> > >
> > > Secondly and more importantly, Sec 5 (Pg 6) in the ICLR version of [32] under the ‘Hyperparameter search’ subsection states that: “**`*All hyperparameters are optimized anew for each algorithm and test domain*`, including hyperparameters like learning rates which are common to multiple algorithms.**” Similarly, Sec 5 (Pg 7) in the arxiv version of [32] under the ‘Hyperparameter search’ subsection states that: "**For each algorithm and test environment, we conduct a random search of 20 trials over the hyperparameter distribution (see Appendix D). We use each model selection method from Section 3 to `*select amongst the 20 models*` from the random search.**” Thus, [32] *DOES NOT* average performance across the 20 trials, but suggests selecting the best hyperparameters among the 20 trials, each of which is obtained from random search. We exactly follow the same procedure in our work too (note that the list of hyperparameters in our rebuttal was randomly selected).
> > >
> > > Also, in the official code implementation of [32] (code link provided in Sec 4, pg 5, of both ICLR and arXiv versions), note that in Lines 114 and Lines 123-128 of `list_top_hparams.py` file , dataset is passed as an argument, which is then used to list the best hyperparameters for each dataset individually (see Line 141). Line 141 in turn calls `model_selection.py`, where Lines 27-40 clearly states that the hyperparameters are chosen per dataset.
> > >
> > > Following this, we use the same procedure and report the top hyper-parameters for each dataset in Table A1 in the Appendix.
> > >
> > > We hope this can clarify our choice. If there is any more information required, please let us know.

---

> > > > ### Comment · Reviewer_zZgJ · 2023-08-15
> > > > **Thanks for the clarification**
> > > >
> > > > My apology for misunderstanding the part. My main concern is resolved now. Perhaps the author can include a representation saying that the model selection method is "test-domain validation" (given that "training-domain validation" is more common in recent works). I'm raising my rating to 5 (not higher because the introduction and experimental parts may require major revisions).

---

> > > > > ### Author Response · Authors · 2023-08-17
> > > > >
> > > > > We thank the reviewer for increasing the rating and are glad to know that our response helped resolve their main concern.
> > > > >
> > > > > We understand the editorial concerns. We will leverage the additional page that NeurIPS provides in the final version to include the clarifications for these editorial concerns, as well as other additional information provided in the rebuttals. In particular, we list below the specific changes we plan to make to the paper:
> > > > >
> > > > > 1. In Sec 1 (L37-41), we will rephrase our claim on motivating our method using a theoretical framework and instead motivate the proposed work as an adversarial DG method based on margin loss because of its advantages as discussed in Sec 4 (L129-144). We will mention our theoretical analysis as an additional dimension of our efforts in this work.
> > > > > 2. In Sec 2 (L65-66 and L73-76), we will describe in detail the existing adversarial methods [38-42] and clearly differentiate the advantage of using the margin loss in our work. In particular, we will highlight the comparison with $H \Delta H$ discrepancy, and how the margin loss allows us to use the classifier/task in our solution.
> > > > > 3. In Sec 6 (L279-280), we will clearly state that we follow [71] in using 'Test-domain validation' for hyperparameter selection.
> > > > > 4. In Sec A2 (L86-87) in the Appendix, we will describe the procedure for hyperparameter tuning followed in [32] and also include a Table that lists the different search range values for each hyperparameter used in our work.
> > > > > 5. We will explicity state in the Limitations section (Sec A6 in the Appendix) regarding the requirement of domain labels during training, similar to recent work such as [71][88].
> > > > > 6. We will include the additional results included in the rebuttal PDF in the paper.
> > > > > (We also have additional comparisons with SD[b] now, which we could not include in our first rebuttal due to time constraints -- our method outperforms SD, and we will include these too.)
> > > > >
> > > > > We hope the above changes capture the requested clarifications and present our research in the paper appropriately to the community. We also hope this can convince the reviewer on our efforts to clearly address pending issues (which are largely editorial at this time), to improve the paper's presentation and to make it useful. We will be happy to provide any further clarifications if needed.

---

### Official Review · Reviewer_MRUq · 2023-07-06

**Soundness:** 3 good
**Presentation:** 2 fair
**Contribution:** 3 good
**Rating:** 6
**Confidence:** 3

**Summary:**

The paper proposes a new adversarial learning objective using a margin based approach for domain generalization. The goal of domain generalization broadly is to build classifiers that are trained on one or more source domains, and are expected to generalize to an unseen target domain. The key idea is to leverage Margin disparity discrepancy (MDD) which quantifies the level of disagreement between decision boundaries of classifiers using their margins. MDD is used as a proxy to understand generalizability of classifiers in this context across multiple domains. Next, the paper establishes an upper bound on the generalization error on any unseen target domain that is within the convex hull of the source domains and MDD. In other words, the objective is to get the decision boundaries across different source domains to agree as much as possible, while also minimizing the empirical errors on them simultaneously.

**Strengths:**

* The margin perspective for domain generalization is a fresh perspective to this, and the paper takes an interesting approach at using an adversarial learning strategy to optimizing this problem.
* Detailed theoretical setup and formulation, which builds on existing work and setup the formulation for MADG
* Results are impressive — on several benchmarks, the proposed method appears to perform competitively.
* good ablations and analysis of the proposed method.

**Weaknesses:**

* I found the paper to be hard to read in general, at its core, the paper is proposing to model error on the unseen target using a convex hull of the source domains — which is a standard idea in many generalization papers. The novelty, in my opinion, is to use the MDD as an objective for determining discrepancy between domains, and the adversarial game. This can be clarified significantly to make sections 4, 5 more readable. There is too much of notation and terminology that obfuscates the reader from understanding the key contributions of the paper. I am taking the proofs and theorems at face value, and have not verified them.
* As far as i understand, MADG requires to train significantly more models (1 per domain, a feature extractor and the main classifier) in order to compute MDD and perform training. This is vastly more complex than any existing method. It may be that making progress in a hard problem like domain generalization requires this, but i think this trade-off needs to be made more explicit — compute vs generalization performance. Whereas a simple ERM or Mixup have little to no overhead and perform very closely on the metrics considered in the paper.

**Questions:**

See above

**Limitations:**

some what, yes.

---

> ### Author Rebuttal · Authors · 2023-08-09
>
> We thank the reviewer for their valuable comments and suggestions. We respond below to each of their concerns/suggestions.
>
> > I found the paper to be hard to read in general, at its core, the paper is proposing to model error on the unseen target using a convex hull of the source domains — which is a standard idea in many generalization papers. The novelty, in my opinion, is to use the MDD as an objective for determining discrepancy between domains, and the adversarial game. This can be clarified significantly to make sections 4, 5 more readable. There is too much of notation and terminology that obfuscates the reader from understanding the key contributions of the paper. I am taking the proofs and theorems at face value, and have not verified them
> >
> **Response:** Thank you for the suggestion, we will make necessary changes to make sections 4 and 5 more readable.
>
> > As far as i understand, MADG requires to train significantly more models (1 per domain, a feature extractor and the main classifier) in order to compute MDD and perform training. This is vastly more complex than any existing method. It may be that making progress in a hard problem like domain generalization requires this, but i think this trade-off needs to be made more explicit — compute vs generalization performance. Whereas a simple ERM or Mixup have little to no overhead and perform very closely on the metrics considered in the paper.
> >
> **Response:** We agree that the current dataset benchmark in domain generalization, DomainBed [32, ICLR 2021], is a challenging one where one model may not outperform all other models across all the constituent datasets (OfficeHome, PACS, VLCS, TerraIncognita, DomainNet) by a significant margin. Even recent papers, Fish [88, ICLR 2022] and Fishr [71, ICML 2022] showcase small improvements across these datasets. In terms of computational cost, following Eq.(17), while we iteratively compute MDD over all domains, our computational cost during training is similar to other benchmark methods as shown below in Table 1. (To add, even simple methods like ERM and Mixup have running times in similar ranges.)
>
> Table 1: Computation cost (GPU ram occupied [GB] and time per step [s]  during model training) and the average accuracy achieved across all the datasets.
> |Model| GPU RAM occupied (GB) | Time per step (s) | Avg. Accuracy (%)|
> |-------|-------|-------|-------|
> |MADG (ours) | ~10.5 |~1.3 |66.0|
> |Fishr [71] | ~15.7 |~0.6 | 65.7|
> |Fish[88]| ~3.4|~1.2 | 64.8|
> |Mixup [82]| ~8.2 | ~0.4 |65.4|
> |ERM [75] |~8.2 | ~0.4| 65.0|

---

> > ### Author Response · Authors · 2023-08-18
> >
> > Dear reviewer,
> >
> > Thank you for your great efforts in reviewing our paper and offering insightful and constructive comments. The deadline for the author-reviewer discussion is approaching. We wanted to check if our rebuttal addressed your concerns, and would appreciate the opportunity to discuss further with you.
> >
> > Understanding the demands on your schedule, we express our sincere gratitude for your review. Your suggestions/comments will be significantly beneficial to the improvement of our work.

---

> > > ### Comment · Reviewer_MRUq · 2023-08-18
> > > **response**
> > >
> > > Thanks for addressing my concerns. I reiterate my observation earlier that this is a new perspective on the domain generalization problem, and the authors have done a good job of empirical validation across a range of challenging benchmarks. As some of the other reviewers noted, there needs to be a significant update to the writing to make the theory more accessible, and writing more clear for a final publication. I also agree with the concerns that the performance w.r.t more recent baselines like MIRO show little to no improvement, but on the overall, i think this paper is a valuable contribution. I will raise my score to reflect this.

---

> > > > ### Author Response · Authors · 2023-08-20
> > > >
> > > > We thank the reviewer for increasing the score and for appreciating our contributions in this paper.
> > > >
> > > > We understand the editorial concerns. We will *leverage the additional page that NeurIPS provides in the final version* to include the clarifications for these editorial concerns, as well as other additional information provided in the rebuttals. We list below a set of changes we plan to make to the paper:
> > > > 1. In Sec 1 (L49-53), we will clearly state the novelty of this work. In particular, we will discuss the advantages of using a margin-based discrepancy (MDD) for the DG problem, as elaborated in Sec 4 (L129-144), and discuss the adversarial learning formulation of the proposed MADG algorithm that leverages MDD.
> > > > 2. In Sec 2 (L65-66 and L73-76), we will describe in detail existing adversarial methods [38-42] and clearly differentiate the advantage of using the margin loss in our work. We will highlight the comparison with $H \Delta H$ discrepancy, and how margin loss allows us to use the classifier/task in our solution.
> > > > 3. To make it easier for a reader to follow Sections 4 and 5, in Sec 4, after the introductory paragraph to motivate (margin-based) MDD, we will revise L145-151 with a clear description of the organizational structure of Sec 4 and 5. In particular, we will state: "*In this section, we derive a generalization bound for an unseen domain based on the margin-based MDD loss in the DG setting. To this end, we first show an upper bound on the unlabeled source domain error given other labeled source domains (Lemma 1 and Theorem 1). We then leverage this to develop the upper bound for the error on an unseen domain that is not necessarily a source domain (Lemma 2, Theorem 2 and Corollary 1). We subsequently analyze the upper bound from Corollary 1 using the Rademacher complexity framework and develop our final generalization bound for the unseen target domain in the DG setting (Lemma 3 and Theorem 3) using our magin-based loss. In Section 5, we show the formulation of the proposed adversarial learning algorithm, MADG, motivated by the generalization bound in Section 4, that employs MDD to address the DG problem.*"
> > > >
> > > > 4. In Sec 7, we will add a 'Computational cost' analysis subsection  that explicitly states the tradeoff between the computation and the generalization performance for our model, and empirically show that our computational cost is similar to other DG baselines. We will also include Table 1 from our rebuttal in this discussion.
> > > >
> > > > Regarding comparison with recent baselines, as stated in our first response, DomainBed is a challenging benchmark and we achieve consistent performance on all its constituent datasets as shown using the AD, GD and M metrics. We initially did not have MIRO in our comparisons, since their experimental setting is different from ours (we follow [71] in the experimental settings). However, our comparisons included in the rebuttal PDF, show that we outperform MIRO on our experimental setting. To go further, we also compare our method, MADG, with another recent method, SD [a, Neurips 2021], as suggested by the reviewers and show results in the tables below. The proposed MADG outperforms SD on all the domains across all the four datasets.
> > > >
> > > > We hope the above changes capture the editorial clarifications and present our research in the paper appropriately to the community. We are grateful for all the reviewer's suggestions, which has only improved the paper and its presentation.
> > > >
> > > > We will be happy to provide any further clarifications if needed.
> > > >
> > > > Table 1: Accuracy (%) on OfficeHome dataset
> > > > |Model|Art|Clipart|Product|Real World| Average|
> > > > |-------|--------|-------|-------|------|------|
> > > > |SD[a]|64.8($\pm0.9$)|49.9($\pm1.2$)|75.6($\pm1.3$)|79.1($\pm0.2$)|67.4
> > > > |MADG (ours)| 68.6($\pm0.5$)|55.5($\pm 0.2$)|79.6($\pm0.3$)|81.5($\pm0.4$)|71.3|
> > > >
> > > > Table 2: Accuracy (%) on PACS dataset
> > > > |Model|Art_Painting|Cartoon |Photo |Sketch | Average|
> > > > |-------|--------|-------|------- |-------|------|
> > > > |SD[a]|82.7($\pm0.9$)|77.4($\pm3$)|97.1($\pm0.7$)|68.5($\pm2.4$)|81.4|
> > > > |MADG (ours)|87.8($\pm0.5$)|82.2($\pm0.6$)|97.7($\pm0.3$)|78.3($\pm0.4$)|86.5|
> > > >
> > > > Table 3: Accuracy (%) on VLCS dataset
> > > > |Model|PASCAL|CALTECH|LABELME|SUN| Average|
> > > > |-------|--------|-------|-------|------|------|
> > > > |SD[a]|72.6($1.7$)|91.4($\pm1.7$)|64.1($\pm1.6$)|63.3($\pm1.2$)|72.3|
> > > > |MADG (ours)|77.3 ($\pm0.1$)|98.5($\pm0.2$)|65.8($\pm0.3$)|73.1($\pm0.3$)|78.7|
> > > >
> > > > Table 4: Accuracy (%) on TerraIncognita dataset
> > > > |Model|L100|L46|L43|L38| Average|
> > > > |-------|--------|-------|-------|------|------|
> > > > |SD[a]|39.5 ($\pm8.7$)|41.4 ($\pm1.2$)|49.2 ($\pm1$)|34.1($\pm6.2$)|41.1|
> > > > |MADG (ours)|60.0 ($\pm1.2$)|45.6($\pm0.5$)|57.4($\pm0.3$)|51.8($\pm0.2$)|53.7|
> > > >
> > > > [a] Pezeshki, Mohammad, et al. "Gradient starvation: A learning proclivity in neural networks." Advances in Neural Information Processing Systems 34 (2021): 1256-1272.

---

### Official Review · Reviewer_FJmm · 2023-07-07

**Soundness:** 3 good
**Presentation:** 3 good
**Contribution:** 3 good
**Rating:** 6
**Confidence:** 3

**Summary:**

- The paper presents a margin-loss based analysis of domain generalization.
- First, the paper derives a bound on margin disparity discrepancy (MDD) of any unseen domain within the convex hull of source domains in terms of the margin loss on source domains, ideal margin loss, and max MDD between any two source domains.
- The paper relates this to any unseen domain via projection onto the convex hull of source domains, and an additional factor $\gamma$
- The paper then realizes the bound for the empirical setting using a bound based on the Rademacher complexity of the function class
- Based on this bound, the paper proposes an adversarial learning method to regularize training by minimizing MDD in addition to classification loss.
- The paper demonstrates the efficacy of the method on a variety of real world datasets in the DomainBed benchmark.

**Strengths:**

1. The paper proposes a principled method for domain generalization, and the empirical method follows neatly from the derived generalization bound.
2. The proposed bound for error on an unseen domain based on margin disparities between source domains is novel to my knowledge.
3. Empirical results show modest improvements over ERM.
4. The paper presents empirical ablations for some design choices used in the algorithm.

**Weaknesses:**

It is not clear if the single optimization step for the adversarial models f’ is sufficient for tightly approximating MDD. The method effectively regularizes a lower bound on the MDD term, which requires the adversarial models to be effective maximizers — this may require many steps for the adversarial models per main-model step.

Table 1 presents results using the model selection strategy of [71], which uses out-of-distribution data as the validation set for picking hyperparameters. Using labelled OOD data for hyperparameter selection limits the conclusions we can draw about if the proposed method works in practice, where we do not usually have access to labelled test data. While the AD, GD, and M metrics allay some of this concern, presenting results with some methodology that could be used for model-selection in practice, such as leave-one-out domain validation would make the empirical results stronger.

**Questions:**

Why is only the feature extractor trained to minimize MDD loss and not the classifier (line 261)?

Theorem 2 shows that the error of any unseen domain can be bounded via a projection onto the convex hull of the source domain, by adding in an additional factor $\gamma$. It is not clear if this factor is small or very large for domain generalization benchmarks. Is there intuition for “how far” an unseen domain in a benchmark is to the source domains?

**Limitations:**

It is not completely clear how significant the empirical results are. The method slightly outperforms ERM on average, but ERM outperforms the method on 2 out of 5 considered datasets, and is competitive on 2 others. Additionally, the reliance on labelled OOD data for model selection limits the conclusions one can draw about the efficacy of the method on the DomainBed benchmark.

---

> ### Author Rebuttal · Authors · 2023-08-09
>
> We thank the reviewer for the valuable comments and suggestions. We respond below to each of the concerns/suggestions.
>
> > It is not clear if the single optimization step for the adversarial models f’ is sufficient for tightly approximating MDD....this may require many steps for the adversarial models per main-model step.
> >
> **Response:** As discussed in Algorithm 1, each classifier f' (Adversarial model) along with the feature extractor is updated once per main-model step. We had found this to be sufficient in our empirical studies. To however answer this further, we report results on the Art domain in OfficeHome dataset with additional adversarial updates per main-model step (t) in Table 1 below. (We had observed similar trends in our earlier studies.) As in the table, adversarial update with t=3 achieves only marginally better performance for the added computational cost.
>
> Table 1: Accuracy (%) for Art domain for different adversarial updates per main-model step
> |\# Adversarial updates| Art|
> |--|--|
> |t=1| 68.6 |
> |t=3|68.7|
>
> >Table 1 presents results using the model selection strategy of [71], which uses OOD data as validation set for picking hyperparameters. Using labelled OOD data for hyperparameter selection limits the conclusions ..... While D, GD, and M metrics allay some of this concern, presenting results with some methodology that could be used for model-selection in practice would make empirical results stronger.
>
> **Response:** As stated in Appendix A2 (L85-86), we follow the recent state-of-the-art work [71] (also supported by recent papers [refs A1-A3]) in using the 'Test-domain' method for hyperparameter selection - primarily for fair comparison of results. (We also reported train-domain results in the Appendix, although that was not our focus in this work.)
> These recent efforts also discuss reasons for this choice, such as avoiding over-reliance on spurious correlations [71, A2] and the underspecification issue as a key reason for failure of deployed ML systems [A3]. Also, [71] argues that it is more realistic that a user will label a few target samples to validate their model's generalization performance before deploying it in practice. We followed these recent efforts in using the test-domain method.
>
> > Why is only the feature extractor trained to minimize MDD loss and not classifier (line 261)?
> >
> **Response:** As seen in Eqns (5) and(6), MDD is defined as the supremum over only the $f'$ scoring function  while the other function $f$ is fixed. Thus, following this definition, our final optimization problem, Eq (22), maximizes MDD in terms of the $f'$ classifier and minimizes MDD in terms of the feature extractor ($\mathcal{G}$). The classifier $f$ is fixed when optimizing the MDD loss.
>
> >Theorem 2 shows that error of any unseen domain can be bounded via a projection onto the convex hull of the source domain, by adding an additional factor $\gamma$. It is not clear if this factor is small or large for DG benchmarks. Is there intuition for “how far” an unseen domain in a benchmark is to the source domains?
> >
> **Response:** As stated in Theorem 2 and Remark 3, $\gamma$ is defined as the projection of the unseen domain on to the convex hull. It is also defined in Thm 2 as the MDD between the convex hull and the unseen domain, which can be further approximated with a combination of two different cross entropy functions as in Eq (20). This equation is equivalent to a $\hat{\rho}$ balanced Jensen-Shannon (JS) Divergence as shown in Propn D.1. in [37]. Thus we can get an approximation of this projection by computing the JS divergence between the two domains (distributions). To answer this better, we studied this paiwise JS divergence between two domains in the OfficeHome dataset, as shown in Table 1 below. Such an analysis provides us with an intuitive value for $\gamma$ by summing the values across the rows in Table 1. We can see from Table 1 that the unseen accuracy is low when the $\gamma$ value is high (expectedly). This is because higher $\gamma$ values incidcate that the unseen domain is far away from the convex hull and hence harder to generalize.  We can see a similar trend for PACS dataset in Table 2 below.
>
> Table 1: Pairwise JS divergence for the OfficHome dataset and the approximated projection value ($\gamma$) of the unseen domain on the convex hull.
> |OfficeHome|Art|Clipart|Product|Real World| ~$\gamma$|Accuracy(%)|
> |-----|-----|-----|-----|-----|-----|-----|
> |Art|0|0.36 |0.35|0.09|0.8|68.6|
> |Clipart|0.36|0|0.31|0.41|1.1|55.5|
> |Product|0.35|0.31|0|0.37|1.0|79.6|
> |Real World|0.09|0.41|0.37|0|0.87|81.5|
>
> Table 2: Pairwise JS divergence for the PACS dataset and the approximated projection value ($\gamma$) of the unseen domain on the convex hull.
> |PACS|Art_painting|Cartoon|Photo|Sketch| ~$\gamma$|Accuracy(%)|
> |-----|-----|-----|-----|-----|-----|-----|
> |Art_painting|0|0.34 |0.08|0.72|1.18|87.8|
> |Cartoon|0.34|0|0.4|0.41|1.24|55.5|
> |Photo|0.08|0.4|0|0.74|1.21|79.6|
> |Sketch|0.72|0.46|0.74|0|1.92|81.5|
>
> >It is not completely clear how significant the empirical results are....
> >
> **Response:** We understand the concern. The current dataset benchmark in domain generalization, DomainBed [32, ICLR 2021], is a challenging one where one model may not outperform all other models across all the constituent datasets (OfficeHome, PACS, VLCS, TerraIncognita, DomainNet) by a significant margin. Even recent papers, SAND-mask [89, ICML 2021 workshop], Fish [88, ICLR 2022] and Fishr [71, ICML 2022] showcase small improvements across these datasets. To further analyze a model’s effectiveness and consistency across these datasets, we evaluate each model with three other metrics (AD, GD, M) as discussed in Sec 6. The proposed model outperforms other models on all three metrics along with the average accuracy metric. As discussed in Appendix A2 (L85-86), we follow recent state-of-the-art work [71] (also supported by recent papers [A1-A3]) for our experimental settings, for fair comparison.

---

> > ### Author Response · Authors · 2023-08-18
> >
> > Dear reviewer,
> >
> > Thank you for your great efforts in reviewing our paper and offering insightful and constructive comments. The deadline for the author-reviewer discussion is approaching. We wanted to check if our rebuttal addressed your concerns, and would appreciate the opportunity to discuss further with you.
> >
> > Understanding the demands on your schedule, we express our sincere gratitude for your review. Your suggestions/comments will be significantly beneficial to the improvement of our work.

---

> > ### Comment · Reviewer_FJmm · 2023-08-19
> > **Response**
> >
> > I thank the authors for addressing my questions. The new ablation resolves my concern related to adversarial optimization.
> >
> > I am currently inclined to retain my score due to the following remaining concerns:
> > 1. I thank the authors for checking the approximate values of gamma on DomainBed. It appears that the empirically observed values are all close to or above 1. Does this not make the bounds of theorems 2 and 3 very loose for settings such as DomainBed where there may be very few source domains (the error rate can be 1 at most anyway)? If that is the case, it would support the story of the paper to motivate why optimizing the other terms still leads to improved DG. Alternatively, are there empirical settings the authors can identify where gamma is small?
> > 2. I appreciate that DomainBed is a challenging benchmark to consistently outperform ERM on and no DG methods lead to meaningful performance improvements. However, to show that MADG is a useful algorithm for domain generalization, there needs to be an experiment where MADG leads to convincingly better results than ERM. This doesn't have to be DomainBed, it could be a simpler synthetic setting (for example with the unseen domain close to or within the convex hull of source domains) — [71] considers ColoredMNIST and a linear task where Fishr outperforms ERM significantly.

---

> > > ### Author Response · Authors · 2023-08-21
> > >
> > > We thank the reviewer for acknowledging our response, and are glad to know that it helped resolve concerns.
> > >
> > > The reviewer is absolutely correct. The empirically observed values of gamma are all close to or above 1 for a challenging benchmark like DomainBed. Hence, in this work, we focus on optimizing the first two terms in Theorem 3 in our proposed MADG method and demonstrate improved performance on the DomainBed benchmark. Optimizing all the terms in Theorem 3 would be a natural and interesting extension of our work. Based on the reviewer's suggestions, we ran experiments on ColoredMNIST (more discussion below), where we noted the gamma values to be small for all domains (expectedly so since the distributions of different domains in ColoredMNIST are relatively close to each other, when compared to real-world datasets).
> > >
> > > Regarding the comparision with ERM model, we thank the reviewer for the suggestion. We conducted studies on the ColoredMNIST dataset, as shown in the tables below. As seen in Table 2, with almost no hyperparamter tuning (due to limited time), the proposed MADG algorithm achieves 65.6\% average accuracy, which is significantly higher than the ERM model (57.8\% average accuracy), thus showing that using a margin-based DG algorithm, MADG, learns better domain-invariant features  on the ColoredMNIST dataset. Besides, Table 1 shows that gamma is small across domains in this dataset, showcasing the promise of the proposed method when the unseen domain is within the convex hull of source domain. We will leverage the additional page provided in NeurIPS for the final version to include these results and discussions.
> > >
> > > We hope that this clarifies the reviewer's concerns. We are grateful for the opportunity to engage with the reviewer, and improve our paper. We'd be happy to provide any further information or clarifications if required and permitted.
> > >
> > > Table 1: Pairwise JS divergence for the ColoredMNIST dataset and the approximated projection value ($\gamma$) of the unseen domain on the convex hull.
> > > |OfficeHome|+90\%|+80\%|-90\%| ~$\gamma$|
> > > |-----|-----|-----|-----|-----|
> > > |+90\%|0|0.019|0.024|0.043|
> > > |+80\%|0.019|0|0.029|0.048|
> > > |-90\%|0.024|0.029|0|0.054|
> > >
> > > Table 2: Accuracy (%) on ColoredMNIST dataset
> > > |Model|+90\%|+80\%|-90\%|Average|
> > > |-------|--------|-------|-------|------|
> > > |ERM|71.8 ($\pm0.4$)| 72.9($\pm0.1$)| 28.7 ($\pm0.5$)| 57.8|
> > > |MADG| 72.3 ($\pm0.3$)| 74.0 ($\pm0.2$)  | 50.5 ($\pm0.4$)|65.6|

---

> > > > ### Comment · Reviewer_FJmm · 2023-08-21
> > > > **Response**
> > > >
> > > > Thanks to the authors for the follow up. The new results on Colored-MNIST provide early evidence that MADG can outperform ERM in settings where the unseen domain is close to the convex hull of the source domains. I have raised my score to reflect these results, and encourage the author to revise the paper to include them.

---

> > > > > ### Author Response · Authors · 2023-08-21
> > > > >
> > > > > We thank the reviewer for increasing the score. We will leverage the additional page provided in NeurIPS for the final version to include these results and discussions.

---

### Official Review · Reviewer_wf8A · 2023-07-07

**Soundness:** 3 good
**Presentation:** 3 good
**Contribution:** 3 good
**Rating:** 5
**Confidence:** 2

**Summary:**

This paper is commendable for its innovative use of a margin-based theoretical framework to solve domain generalization problems, which contrasts with the largely heuristic and empirical approaches adopted by existing methods. By grounding their approach in a theoretical foundation, the authors provide more interpretable solutions that can contribute to the field of domain generalization.

**Strengths:**

- The claim presented in the paper is strongly supported by theoretical analysis, as well as the extensive ablation studies presented in section 7. This speaks to the methodological rigour of the study and strengthens its credibility.
- The development of a theoretical framework for solving domain generalization problems is indeed a significant strength of this paper. This approach not only enhances the comprehensibility and reproducibility of the method, but also contributes to the broader understanding of the problem.

**Weaknesses:**

- While the average performance of the proposed method is the highest, the performance gap across different tasks is quite significant. The method shows the best performance on the Office Home dataset, but this alone is not enough to convincingly demonstrate the overall superiority of the method. Therefore, the claimed significance of the proposed method appears to be overstated.
- The authors' claim that a theoretical approach is necessary is quite a strong statement that seems to undermine the importance of empirically validated methods in the field. This makes it hard to fully agree with their motivation.
- While the authors make a reasonable argument for the use of margin as a metric in generalization, they do not provide a sufficient justification for its relevance in domain generalization problems, particularly where style shifts are involved. This makes it difficult to understand the authors' motivation for their margin-based approach.
- Similarly, the reasoning behind the use of adversarial learning (min-max framework) is not clearly explained. The proposed method appears to be similar to the one used in [1], and a clear explanation of the differences between these two methods, apart from the task, would be beneficial.
- The paper lacks a comparison with recent Domain Generalization (DG) works [2-4]. This makes it difficult to assess how the proposed method stacks up against the state of the art.

[1] Maximum Mean Discrepancy Test is Aware of Adversarial Attacks

[2] Fine-Tuning can Distort Pretrained Features and Underperform Out-of-Distribution

[3] Domain Generalization by Mutual-Information Regularization with Pre-trained Models

[4] SIMPLE: Specialized Model-Sample Matching for Domain Generalization

**Questions:**

Please see weakness

**Limitations:**

The authors have well described their limitations.

---

> ### Author Rebuttal · Authors · 2023-08-09
>
> We thank the reviewer for the valuable comments and suggestions. We respond below to each of the concerns/suggestions.
>
> > While the average performance of the proposed method is the highest, the performance gap across different tasks is quite significant.....
>
> **Response:** We agree and in Sec 6 (L288- 292), we discuss this concern and in order to reward model's consistent performance across datasets, we include three other metrics (AD, GD, M), discussed in Sec 6, which reward consistent performance across benchmarks. As seen from Table 1 in main paper, we outperform all earlier methods on these three metrics and the average accuracy metric. Besides, the current dataset benchmark in domain generalization, DomainBed [32, ICLR 2021], is a challenging one where one model may not outperform all other models across all the constituent datasets (OfficeHome, PACS, VLCS, TerraIncognita, DomainNet) by a significant margin. Even recent papers, SAND-mask [89, ICML 2021 workshop], Fish [88, ICLR 2022] and Fishr [71, ICML 2022] showcase similar improvements across these datasets.
>
> > The authors' claim that a theoretical approach is necessary is quite a strong statement that seems to undermine the importance of empirically validated methods in the field....
>
> **Response:** In hindsight, we agree with this observation, and thank you for pointing this out. We will revisit the phrasage and temper our claim. Our overall intention was to suggest that having strong theoretical underpinnings along with improved performance is an added advantage for a method. As discussed in Sec 1 (L39-42), our motivation was to develop an adversarial DG algorithm that uses a margin-based discrepancy because of its advantages, discussed in Sec 4. Such an approach has not been done for DG before. We also develop a generalization bound for unseen domain based on margin loss as they are more informative than 0-1 loss based bounds.
>
> > While the authors make a reasonable argument for the use of margin as a metric in generalization, they do not provide a sufficient justification for its relevance in DG problems, particularly where style shifts are involved....
> >
> **Response:** We apologize if this did not come through clearly. As stated in Sec. 4 (L136- 139), one advantage of using margin-based discrepancy (MDD) is that it considers the task (classifier) in its formulation as shown in Eq (5). MDD learns invariant features across domains based on the idea of a bi-classifier prediction. Let's consider the MDD between the two domains ($D_{s_i}$,$D_{s_k}$) as shown in Eqns (5)&(22) (consider the case where $N_s=2$ and $j=1$). We have two different classifiers (scoring functions) that take input from the feature extractor and try to classify $D_{s_i}$ samples correctly, whereas for $D_{s_k}$ samples only classifier $f$ tries to classify it correctly. Simultaneously the two classifiers are also trained to detect $D_{s_k}$ samples far from $D_{s_i}$'s support. This is possible because $D_{s_k}$ samples far from $D_{s_i}$’s support cant be easily discriminated by $f'$ classifier  The feature extractor then tries to generate $D_{s_k}$ features such that they are close to the support of $D_{s_i}$. In this way, we align the domains and learn domain-invariant features. We will clarify this.
>
> > Similarly, the reasoning behind the use of adversarial learning (min-max framework) is not clearly explained. ....a clear explanation of the differences between this and the one used in [1], apart from the task, would be beneficial.
> >
> **Response:** As discussed in Sec 5 (L243-251), we formulate our objective as a minimization problem as shown in Eq (18). We then expand the second term (Empiricial MDD) in this problem using Eq (6). As empirical MDD is defined as the supremum over $f'$ (maximization), the overall optimization now becomes a minimax game. We hence use adversarial learning to solve our objective function. We will explain this clearly.
>
> We note that our discrepancy, MDD, is different from the more common MMD. The paper [1] uses Maximum Mean Discrepancy (MMD), whereas, in our paper, we use Margin Disparity Discrepancy (MDD), which is fundamentally different from MMD formulation as shown in Eqns (5) and (6).
>
> >The paper lacks a comparison with recent DG works [2-4]. This makes it difficult to assess how the proposed method stacks up against the state of the art.
> >
> **Response:** As in Sec 6 (L278-280) we evaluate our model on five different datasets from DomainBed benchmark [32]. To the best of our knowledge, this is fairly comprehensive, when compared to most recent efforts. To answer this further, [2] evaluates only on the Domainnet dataset of the DomainBed benchmark, and also uses a smaller version of this dataset (3 instead of 6 domains, and 40 classes instead of 345). Note that DomainBed consists of 5 datasets (OfficeHome, PACS, VLCS, TerraIncognita, DomainNet). To provide a comparison with [2], we train our model on their version of the domainnet dataset. Table 1 in attached rebuttal.pdf shows the results. We see that our model outperforms LP-FT [2] on all three domains. As stated in Appendix A2 (L85-86), we follow recent state-of-the-art work [71] (also supported by recent papers [A1-A3]) in using the `Test-domain` methodology for hyperparameter selection. [3] proposes MIRO model and evaluates only for  a training-domain model selection method. For fair comparision, we re-run the MIRO model for three trials on our setting. From Table 2 in rebuttal.pdf, we see that MADG outperforms MIRO on two domains and achieves higher average accuracy. The paper [4] proposes an ensemble algorithm (collection of multiple methods) for DG. Our method is a non-ensemble one, and can be a part of their method by itself.

---

> > ### Comment · Reviewer_wf8A · 2023-08-14
> >
> > Thank you to the authors for their comprehensive response and the additional experiments provided. Concerns about 2-4 have been resolved. However, I still have concerns regarding the performance gap evaluations across different tasks and the comparisons with recent studies. From the response provided in the PDF, I am not entirely convinced that MADG surpasses MIRO or even when compared to Fishr. Consequently, due to limited evaluation, I will maintain my score at 5.

---

> > > ### Author Response · Authors · 2023-08-20
> > >
> > > We thank the reviewer for acknowledging our response, and are glad to know that it helped resolve concerns.
> > >
> > > Considering the experimental studies in the main paper, ablation studies, additional results on other experimental settings in the appendix and those in the rebuttal, we humbly submit that our evaluations are fairly comprehensive. We believe that the reviewer is pointing to the performance comparisons here than the amount of experimental studies itself.
> > >
> > > Regarding the gap in performance evaluation, as stated in our first response, DomainBed is a challenging benchmark and we achieve consistent performance on all its constituent datasets as shown using the AD, GD and M metrics. We initially did not have MIRO in our comparisons, since their experimental setting is different from ours (we follow [71] in the experimental settings). However, our comparisons included in the rebuttal PDF, show that we outperform MIRO on our experimental setting. To go further, we also compared our method, MADG, with another recent method, SD [a, Neurips 2021], as suggested by the reviewers and show results in the tables below. The proposed MADG outperforms SD on all domains across all the four datasets. We will add these to the paper.
> > >
> > > Regarding performance comparison with Fishr, as seen in Table 1 of our paper, our method outperforms the Fishr model on all three evaluation metrics that reward consistency of performance across domains/datasets: AD, GD, M.
> > >
> > > We'd like to add here that existing work on DG are designed from different perspectives, such as domain-invariant representation learning, data/feature manipulation, and meta-learning/optimization. We believe each perspective has its specific advantages and can achieve superior performance on specific domains/datasets. Consequently, it may not be easy for a method from one perspective to achieve a large improvement than another perspective across domains/datasets. We show that our method outperforms other methods that use the same experimental setting (following [71]) consistently. Our method has a novel contribution in its margin-based discrepancy and adversarial formulation. The combination of our method with complementary ones, e.g. those based on data/feature augmentation may only provide opportunities for future work and even better results.
> > >
> > > We hope this can address any pending concerns w.r.t. performance evaluation. We are grateful for the opportunity to engage with the reviewer, which has only helped us improve the paper. If there is any more information required, please let us know.
> > >
> > >
> > > Table 1: Accuracy (%) on OfficeHome dataset
> > > |Model|Art|Clipart|Product|Real World| Average|
> > > |-------|--------|-------|-------|------|------|
> > > |SD[a]|64.8($\pm0.9$)|49.9($\pm1.2$)|75.6($\pm1.3$)|79.1($\pm0.2$)|67.4
> > > |MADG (ours)| 68.6($\pm0.5$)|55.5($\pm 0.2$)|79.6($\pm0.3$)|81.5($\pm0.4$)|71.3|
> > >
> > > Table 2: Accuracy (%) on PACS dataset
> > > |Model|Art_Painting|Cartoon |Photo |Sketch | Average|
> > > |-------|--------|-------|------- |-------|------|
> > > |SD[a]|82.7($\pm0.9$)|77.4($\pm3$)|97.1($\pm0.7$)|68.5($\pm2.4$)|81.4|
> > > |MADG (ours)|87.8($\pm0.5$)|82.2($\pm0.6$)|97.7($\pm0.3$)|78.3($\pm0.4$)|86.5|
> > >
> > > Table 3: Accuracy (%) on VLCS dataset
> > > |Model|PASCAL|CALTECH|LABELME|SUN| Average|
> > > |-------|--------|-------|-------|------|------|
> > > |SD[a]|72.6($1.7$)|91.4($\pm1.7$)|64.1($\pm1.6$)|63.3($\pm1.2$)|72.3|
> > > |MADG (ours)|77.3 ($\pm0.1$)|98.5($\pm0.2$)|65.8($\pm0.3$)|73.1($\pm0.3$)|78.7|
> > >
> > > Table 4: Accuracy (%) on TerraIncognita dataset
> > > |Model|L100|L46|L43|L38| Average|
> > > |-------|--------|-------|-------|------|------|
> > > |SD[a]|39.5 ($\pm8.7$)|41.4 ($\pm1.2$)|49.2 ($\pm1$)|34.1($\pm6.2$)|41.1|
> > > |MADG (ours)|60.0 ($\pm1.2$)|45.6($\pm0.5$)|57.4($\pm0.3$)|51.8($\pm0.2$)|53.7|
> > >
> > > [a] Pezeshki, Mohammad, et al. "Gradient starvation: A learning proclivity in neural networks." Advances in Neural Information Processing Systems 34 (2021): 1256-1272.

---

### Official Review · Reviewer_u5h1 · 2023-07-07

**Soundness:** 3 good
**Presentation:** 3 good
**Contribution:** 3 good
**Rating:** 5
**Confidence:** 4

**Summary:**

This paper investigates domain generalization (DG) using a margin-based theoretical framework. The authors first formulate the generalization upper bound by leveraging the margin disparity discrepancy (MDD). Then, an adversarial learning strategy (MADG) is devised to minimize the empirical MMD between source domains. The experiments on the DomainBed benchmark further demonstrate that the proposed MADG could outperform the current state-of-the-art methods.

**Strengths:**

1. The paper is clearly written, and the method is well-designed.
2. The theory of margin-based generalization bound is interesting and could benefit the further designs of DG methods.
3. The proposed MADG could outperform recent state-of-the-art methods.


**Weaknesses:**

1. The primary concern of the reviewer is in its effectiveness. While MADG outperforms most baselines in the DomainBed benchmark, the improvement is marginal/minor. This made it unclear to me if the method actually works better, or it is just a product of optimizing some hyperparameters.
2. The motivation, “very little work has been done in developing DG algorithms that are well-motivated by theoretical”, lacks soundness. In fact, there have been notable works in the literature that utilize theoretical frameworks to analyze the generalization problem, such as gradient matching [71, 88] and invariant risk minimization [80]. It could be beneficial if the authors could provide a detailed comparison with such theoretical methods.
3. In addition to the accuracy comparison, could the authors provide a more in-depth analysis regarding the training dynamics? This would help shed light on the inner workings of the MADG approach.
4. Many advantages, such as efficient optimization, stated in the abstract are not well verified. Specifically, following Eq. (17), it seems that the estimation process could introduce a significant computational cost, as it iteratively computes for all domains.


**Questions:**

1. Table 1 could be simplified, as many baselines are unnecessary.
2. The common approach to compare DG methods is the training-domain validation. Why do the authors mainly utilize the test-domain selection?


**Limitations:**

See above comments.

---

> ### Author Rebuttal · Authors · 2023-08-09
>
> We thank the reviewer for the valuable comments and suggestions. We respond below to each of the concerns/suggestions.
> > 1. The primary concern of the reviewer is in its effectiveness. While MADG outperforms most baselines in the DomainBed benchmark, the improvement is marginal/minor. This made it unclear to me if the method actually works better, or it is just a product of optimizing some hyperparameters.
> >
> **Response:** We understand the concern. The current dataset benchmark in domain generalization, DomainBed [32, ICLR 2021], is a challenging one where one model may not outperform all other models across all the constituent datasets (OfficeHome, PACS, VLCS, TerraIncognita, DomainNet) by a significant margin. Even recent papers, SAND-mask [89, ICML 2021 workshop], Fish [88, ICLR 2022] and Fishr [71, ICML 2022] showcase small improvements across these datasets. To further analyze a model’s effectiveness and consistency across these datasets, we evaluate each model with three other metrics (AD, GD, M) as discussed in Sec 6. The proposed model outperforms other models on all three metrics along with the average accuracy metric.
> > 2. The motivation, “very little work has been done in developing DG algorithms that are well-motivated by theoretical”, lacks soundness. In fact, there have been notable works in the literature that utilize theoretical frameworks to analyze the generalization problem, such as gradient matching [71, 88] and invariant risk minimization [80]. It could be beneficial if the authors could provide a detailed comparison with such theoretical methods.
> >
> **Response:** We apologize for not being clearer on this. The gradient matching papers [71,88] propose the idea of learning invariant features by domain-level gradient matching; in particular, [71] use gradient variance matching, and [88] use gradient mean matching. These papers don't develop generalization bounds for the unseen domain; instead, they theoretically analyze the inconsistency score function to find optimal weights across all domains. Even though [80] develops a generalization bound for the unseen domain, it uses a different causal mechanism-based approach. In this work, we present a new perspective to the problem based on margin-based bounds and adversarial learning, which considers convexity relations between domains and the alignment and discrepancy between these domains. We will revisit the phraseage as suggested.
>
> > 3. In addition to the accuracy comparison, could the authors provide a more in-depth analysis regarding the training dynamics? This would help shed light on the inner workings of the MADG approach.
> >
> **Response:** The key aspects of training dynamics have been analyzed in Sec. 5 (L252-258) and Eq. (20) the optimization problem consists of two loss terms, Classification and Transfer Loss, that are updated in an two-step training methodology. Fig. 1 and 2 in rebuttal.pdf shows the classification and transfer loss plots for the Art and Product domain of OfficeHome.
>
> > 4. Many advantages, such as efficient optimization, stated in the abstract are not well verified. Specifically,.. Eq. (17), it seems that the estimation process could introduce a significant computational cost, as it iteratively computes for all domains.
> >
> **Response:** As seen from Eqns (5) and (6), the MDD between two domains is defined as the supremum over only one variable (f\`) as opposed to other discrepancies used in related work such as H$\Delta$H, which involves a supremum over two variables. This makes MDD relatively more efficient to optimize when compared to other discrepancy measures; we will clarify this. In terms of computational cost, following Eq.(17), while we iteratively compute MDD over all domains, our computational cost during training is similar to other benchmark methods as shown below in Table 1 in rebuttal.pdf. (To add, even simple methods like ERM and Mixup have running times in similar ranges.)
>
> > Table 1 could be simplified, as many baselines are unnecessary.
> >
> **Response**: Thank you for the suggestion, we will simplify the table by either dividing it into sections or moving some results to the appendix.
>
> > The common approach to compare DG methods is the training-domain validation. Why do the authors mainly utilize the test-domain selection?
> >
> **Response:** As stated in Appendix A2 (L85-86), we follow recent state-of-the-art work [71] (also supported by recent papers [refs A1-A3]) in using the 'Test-domain' method for hyperparameter selection. These recent efforts also discuss reasons for this choice: (i) [71] argues that learning causal mechanisms is not useful in the Training-domain method, esp when correlations are more predictive in training than causal features. The paper suggests that using 'Test-domain' selection is more realistic since a user will easily label a few target samples to validate a model's generalization performance before deploying it; (ii) [A1] states that In-domain (ID) validation sets (Training-domain) eliminates the advantages of using DG-specific models over ERM models, as ID accuracy is often at odds with generalization accuracy. In our work, restricting the $\rho$ value selection using ID validation can hurt the model's generalization performance as shown in the relationship between margin and generalization in Thm 3; (iii) [A2] argues that high ID accuracy can be achieved by depending on spurious patterns and is not indicative of generalization performance; (iv) [A3] states that Training-domain method suffers from underspecifications as a reason for the failure of ML systems deployed in the real world. It defines underspecification as a scenario where distinct predictors, primarily validated on Training-domain method, report equivalently strong held-out performance but behave differently in test. We follow these recent efforts in using the test-domain method. (We also reported train-domain results in the Appendix, although that was not our focus in this work.)

---

> > ### Comment · Reviewer_u5h1 · 2023-08-16
> >
> > The authors have addressed the concerns, in particular regarding the validations.

---

> > > ### Author Response · Authors · 2023-08-17
> > >
> > > We are happy to know that the reviewer's concerns have been addressed. We sincerely appreciate the thoughtful feedback which helped us improve the presentation of our work.  We will leverage the additional page that NeurIPS provides in the final version to include the clarifications for the reviewer's concerns, as well as other additional information provided in the rebuttals.
> > >
> > > We would appreciate if you could kindly consider revising the score accordingly. Once more, we express our gratitude for the time you've dedicated to these thoughtful discussions.

---

### Author Rebuttal · Authors · 2023-08-09

We thank all reviewers for their positive feedback: proposed method is well designed [Ru5h1, RFjmm]; the method is well supported by theoretical analysis as well as extensive ablation studies [Rwf8A, RFjmm, RMRuq]; the development of a theoretical framework for DG problem not only enhaces the comprehensibility and reproducibility of the method, but also contributes to the broader understanding of the problem [Rwf8A, Ru5h1, RzZgJ] and the results of the proposed MADG model are good [Ru5h1, RMRuq]. We respond to each reviewer's comments below.

---

### Decision · Program_Chairs · 2023-09-21

**Decision:**

Accept (poster)

**Comment:**

The theoretical results of the proposed method for DG are sound, while the experimental results only show minor improvement in performance. The paper may need a major revision in editing to make it easy and clear to follow.

Overall, this is a borderline paper. As most of the major concerns have been addressed after rebuttal, this paper is slightly above the borderline.